# Spatial spillover effect of China's tax and fee reduction policies on independent research and development evidence from dynamic Spatial Dubin analysis

**Meng Wu, Ruoyuan Sun** *

School of Economy, Shandong Women's University, Jinan, China

* 853740844@qq.com

## Abstract

To test the driving effect of China's tax and fee reduction policies on independent innovation, we established a model of Dynamic Spatial Durbin (SDM) and introduced DMSP/OLS night lighting data and Malmquist productivity index for partial differential decomposition. We found that: (1) Affected by the tax and fee reduction policies, the local province tends to increase the level of independent innovation in the short term, while neighboring provinces tend to purchase and rely on foreign technology; (2) In the long term, the tax and fee reduction policies do not significantly increase the level of independent innovation in local and neighboring regions; (3) There is a strategic choice behavior of local government between political promotion incentives and promoting independent innovation; (4) The policy externality of tax reduction and fee reduction has a two-way feedback effect. We conclude that: (1) The spatial agglomeration characteristics of tax and fee reduction policies require the government to fully consider the local innovation and economic foundation, and break the resource endowment of administrative divisions; (2) The spatial feedback feature of the tax and fee reduction policies requires the government to focus on the two-way interaction of independent innovation in the adjacent regions, rather than just one-way assistance, imitation and learning; (3) The spatial lag characteristics of tax and fee reduction policies require the government to establish a accountability system or life-long system for innovative performance evaluation. Moreover, the study fails to provide causality evidence from the spatial agglomeration and spatial time-delay.

## 1. Introduction

Since the 1980s, the developed countries have carried out remarkable tax reform under the leadership of the tax system reform in Britain and the United States, with the main line of reducing tax rates and expanding the tax base. At the end of 2017, the then US President Trump signed the tax reform bill with the general tone of tax reduction, which was the fourth round of tax reduction in the US since 1980. In addition to the United States, the United Kingdom, France, Germany and other countries have also introduced corporate tax

**Funding:** The author(s) received no specific funding for this work.

**Competing interests:** The authors have declared that no competing interests exist.

reduction policies in recent years to promote economic development, stimulate investment and improve the level of scientific and technological progress. The Chinese government has proposed a supply side structural reform driven by innovation and interpreted the new development concept as "innovation, coordination, green, openness, and sharing". Among them, innovation is prioritized and regarded as the eternal driving force for national development [1].

China's technological innovation has gone through four stages: the beginning of technology introduction and absorption, the acceleration of technology introduction and absorption, independent innovation, and innovation-driven development. Among them, according to utilization of primary and secondary technological innovation, independent research and development presents a variety of paths, including "introduction → absorption → innovation", "purchase → transformation → innovation", or "introduction → insufficient absorption → reintroduction", "purchase → insufficient transformation → repurchase".

We have sorted out the relevant policies and measures for reducing taxes and fees. The overall evolution of China's tax and fee reduction policies has gone through three stages: During the period from 2008 to 2011, in response to the impact of the international financial crisis, especially when the export growth rate was negative 16% in 2019, macroeconomic policies were mainly based on demand management measures to respond to business cycle fluctuations. The tax and fee reduction policies during this period were mainly structural tax reductions; From 2012 to 2018, the tax and fee reduction policies, as an important component of the supply side structural reform "cost reduction" measures, played an increasingly important role in active fiscal policies. With the implementation of a series of policies such as the pilot and comprehensive implementation of the "business tax to value-added tax" reform, universal fee reduction, halving of income tax for small and micro enterprises, additional deduction of enterprise research and development expenses, and the combination of comprehensive and classified personal income tax, the scale of tax and fee reductions continues to expand, and the macro tax burden level begins to decline; Since 2019, in order to deal with the trade dispute between China and the United States and the impact of the COVID-19, on the basis of the tax and fee reductions of about 1 trillion yuan and 1.3 trillion yuan in 2017 and 2018, the scale of burden reduction of policies such as the larger scale tax and fee reductions implemented in 2019 and the large-scale rescue and assistance launched against the epidemic in 2020 has reached 2.36 trillion yuan and 2.6 trillion yuan respectively, and the scale of new tax and fee reductions in 2021 is about 1.1 trillion yuan.

Many literatures focus on the innovation practice of the country or the region and study the relationship between scientific and technological progress and economic development. Kaplinsky and Morris argued that under the dynamic role of innovation, it could accelerate the generation of more value added, however focusing on the level of industrial structure upgrading, in a competitive environment, higher innovation efficiency and faster innovation was an important potential to stimulate industrial restructuring and industrial structure upgrading [2]. Through empirical analysis based on 40 industrial production sectors, Kevin argued that innovation from firms changed the output level of industry, while industries with high growth rates possessed higher profits and further contributed to the level of local economic development [3]. Caiani further clarified the role of science and technology innovation for economic development, especially for upgrading the industrial structure, which was prevalent in a variety of industries such as finance as well as chemicals [4]. Restuccia and Rogerson found through related research that better national government policies would be very helpful to regional innovation capacity enhancement, while distorted national government policies would form a more distorted allocation of enterprise factor resources, that caused inter-

regional innovation differences, which in turn, to some extent, hindered local economic growth and inhibited the generation and enhancement of regional innovation activities [5, 6]. Utterback and Abernathy focused on the specific path of the role of innovation in driving industrial restructuring with a micro product production and sales perspective, and on this basis, they proposed a better-known model of industrial upgrading and transformation [7]. But We believe that as a form of promoting innovation, social and economic development is not restricted by administrative regions, which means that independent research and development should not have inherent administrative boundaries, and it often flows through the cross-regional exchange of capital, human resources and advanced technologies [8]. An important feature of economic development is that the improvement of some advanced technologies in an area will make the connection with the surrounding areas through spillover, which is the so-called externality [9]. For example, in the early stage of economic germination, neighboring regions will learn and communicate with each other due to trade exchanges, and there is the diffusion and overflow of production technology or manufacturing process [10]. Due to the political and economic links between different regions, independent R&D presents Externality [11]. This shows that although local administrative divisions are governed by boundaries, the role of innovation and policy effects is not limited to specific geographical boundaries.

On the basis of the above documents, We attempt to explore the relationship between tax and fee reduction policies and independent innovation, and to comprehensively examine the effect relationship between local province and neighboring provinces in the short and long term. Specifically, we use the coefficient partial differential decomposition of the dynamic Space Dubin Model to explore the relationship between tax reduction and innovation. We explained the applicability of spatial metrology in analyzing the policy effect and well-establishing the system of comprehensive evaluation of the policy effect of reducing taxes and fees. In detail, with the help of the agglomeration effect of innovation, we study the heterogeneity of tax and fee reduction policies in promoting innovation in local and nearby regions; With the help of the feedback effect of innovation, we discuss the choice of tax reduction and fee reduction policy for the promotion form of regional independent research and development; With the help of the time lag of innovation, we reveal the realization mode of the promotion effect of tax and fee reductions on innovation. In addition, our research is different from the innovative development model of developed countries such as the United States. For the United States, as a super-developed country, the United States is a pioneer in leading scientific and technological progress. Theoretically, it is the starting point of innovation. As a developing country, it is difficult to become the starting point of innovation without the innovation foundation accumulated by the United States and other superpowers. To achieve a higher level of independent research and development, we must rely on the advantages of domestic policies and the level of foreign technology absorption and introduction. How to promote independent research and development on the basis of tax and fee reduction policies is our research focus. The research has certain reference significance for developing countries to achieve scientific and technological progress and independent research and development. We have combed the development model of China's innovation practice, which can enrich the innovation theory, especially in developing countries, to a certain extent. For example, people can select "introduction → absorption → innovation", "purchase → transformation → innovation ", or "introduction → insufficient absorption → reintroduction", "purchase → insufficient conversion → repurchase". Therefore, whether in theory or practice, our research will help developing countries choose appropriate innovation theories and independent research and development practices, rather than copy the innovation theories and models of developed countries.

## 2. Model establishment and hypothesis deduction

With the help of mathematical model, we used the constant effect function of substitution elasticity (CES) nested in Cobb Douglas (CD) function, and further introduced the elements of innovation and tax reduction policy to derive the research hypothesis through mathematical model deduction.

### 2.1 Partial differential decomposition method and theoretical hypothesis

Obviously, tax and fee reduction policies has significant spillover effects. Based on this, this study introduced to Dixit-Stiglitz monopoly competition model (monopoly competition D-S model), and added tax and fee reduction policies adjustment around the core-edge theory. The core-edge theoretical model is an invariant effect function of substitution elasticity (CES) nested in the Cobb-Douglas (CD) function. On the basis of this framework structure and for the convenience of subsequent mathematical model derivation, the following assumptions are proposed: the first, there are two open regions. The free flow of production factors across borders can be realized, which is the key to explore the influence of externalities; The second, through combing relevant literature, it is found that developed regions have good infrastructure, which can attract more investment and significantly promote technological progress [12]. Therefore, technological progress in developed regions can generate spillover, while relatively poor regions mainly absorb the technology spillover from developed regions as the driving force of their own innovation and development due to the limitation of economic development level.

It is assumed that there are two types of productive economies in the economy, which include labor-intensive and capital-intensive sectors. The production function calculates output in terms of input factors, included labor $L$ and capital $K$. Then it is assumed that the output of labor-intensive sectors is $P_L$, and that of capital-intensive sectors is $P_K$. At this point, the region's gross income can be expressed as $P_Y = F(P_L, P_K)$, and the CES production function after substituting each factor of production is

$$P_Y = \left[ \gamma P_L^{\frac{\kappa-1}{\kappa}} + (1-\gamma) P_K^{\frac{\kappa-1}{\kappa}} \right]^{\frac{\kappa-1}{\kappa}} \tag{1}$$

In the above formula, $\gamma$ and $1 - \gamma$ are the proportion of the contribution of labor and capital to the total output respectively, and $\kappa$ is the elasticity of substitution of the two production factors. In addition, it is assumed that the final output is only related to the factor input and its corresponding technology level, and the technology level of labor-intensive industry is $N_L$, and the technology level of capital-intensive industry is $N_K$, then the following two expressions can be obtained:

$$P_L = N_L f(L, \bar{K}) \tag{2}$$

$$P_K = N_K f(K, \bar{L}) \tag{3}$$

Eqs (2) and (3) describe the maximum benefits that can be obtained by each production department when the input of a certain factor is kept constant. In addition, considering the regions with backward economic development level and assuming that the developed regions have external technology spillovers, the influence coefficient is set as $\beta$. Existing studies have confirmed that the intensity of technology spillover is closely related to the technology gap between rich and poor regions [13]. Obviously, the relationship between the two can be represented as $N_i = \beta N_i'$, $N_i$ refers to the level of technological development of developed areas, $N_i'$

refers to the level of technological progress of backward areas, which can be expressed as follows:

$$P_L = N_L f(L, \bar{K}) = \beta \lambda N'_L f(L', \bar{K}') + \varepsilon \tag{4}$$

$$P_K = N_K f(K, \bar{L}) = \beta \lambda' N'_K f(K', \bar{L}') + \varepsilon' \tag{5}$$

$\lambda$, $\lambda'$ and additional items $\varepsilon$, $\varepsilon'$ are the parameters that distinguish the impact of technological progress such as independent research and development from other factors on the output of two economies. Through the theoretical inference and research hypothesis analysis above, the determinants of technological development level in backward areas can be summarized as follows: First, local fiscal and tax policies. This study focuses on tax reduction policies, assuming tax reduction $T_i$; Second, developed areas of technology spillover $\beta$. Based on the above assumptions, the level of technological progress in economically backward areas can be expressed as:

$$N'_i = \delta_i f(T_i) \left( \frac{N_i}{N'_i} \right)^{\alpha} \tag{6}$$

Considering that local fiscal and tax policies can affect the level of scientific and technological development [14, 15], and no matter whether the government reduces tax on a certain technological research project, the existing level of the technology is not equal to 0, so the function condition in Eq (6) can be set: $f(0) > 0$ and any $T_i \geq 0$, there is $f'(T_i) > 0$. In addition, $N_i/N'_i$ measures the technological difference reflected by two economies at different levels of development, where $\alpha$ indicates the effect intensity of the gap affecting the technology spillover. In general, there is $N_i \geq N'_i$, and $\delta_i$ controls the relevant influencing factor. Therefore, the profit function of the different development sectors can be expressed in the following form:

$$R_L = P_L N_L f(L, \bar{K}) - \mu L - \omega(T'_L) \tag{7}$$

$$R_K = P_K N_K f(K, \bar{L}) - \varphi K - \omega(T'_K) \tag{8}$$

$$R'_L = P'_L N'_L f(L', \bar{K}') - \mu L' - \omega(T_L) \tag{9}$$

$$R'_K = P'_K N'_K f(K', \bar{L}') - \varphi K' - \omega(T_K) \tag{10}$$

The parameters of the above equation are: $P_i$ and $P'_i$. They are the prices of output products of different types (such as labor-intensive and capital-intensive) in the two economies respectively. Where $\mu$ is the wage rate and $\varphi$ is the rent rate. In an economy, that can realize free flow of factors. Although the economic development level of each region is different, the wage rate and the rent rate can eventually be in a stable state of equilibrium. $\omega(x)$ represents the independent research and development costs of different factor production departments under the influence of government tax and fee reduction policies. From what has been discussed above, as far as backward areas are concerned, the first-order partial derivatives of $T_i$ are respectively solved for Eqs (9) and (10) above, and the results are substituted into Eq (6) to obtain:

$$\frac{N'_K}{N'_L} = \left( \frac{K'}{L'} \cdot \frac{P'_K}{P'_L} \cdot \frac{\delta_K}{\delta_L} \right)^{\frac{1}{2}} \left( \frac{N_K}{N_L} \right) \left[ \frac{f'[\eta(T_K)]}{f'[\eta(T_L)]} \cdot \frac{\eta'(T_K)}{\eta'(T_L)} \right]^{\frac{1}{2}} \tag{11}$$

Substitute Eq (2) into the first derivative of Eq (1), and express it as the ratio of price levels of two factors in relatively backward areas, and Eq (12) can be obtained:

$$\frac{P'_K}{P'_L} = \frac{\gamma}{1-\gamma}\left(\frac{N'_K K'}{N'_L L'}\right)^{-\frac{1}{\kappa}} \tag{12}$$

Substituting Eq (12) into Eq (11), we get:

$$\frac{N'_K}{N'_L} = \left(\frac{K'}{L'}\right)^{\frac{\kappa-1}{1+\kappa\alpha}}\left(\frac{\delta_K}{\delta_L}\cdot\frac{\gamma}{1-\gamma}\right)^{\frac{\kappa}{1+\kappa\alpha}}\left(\frac{N_K}{N_L}\right)^{\frac{\kappa\alpha}{1+\kappa\alpha}}\left[\frac{f'[\eta(T_K)]}{f'[\eta(T_L)]}\cdot\frac{\eta'(T_K)}{\eta'(T_L)}\right]^{\frac{1}{\alpha}} \tag{13}$$

Further, Acemoglu proposed that technological development can adjust the marginal output of capital or labor input and influence its selection bias, so as to cause the technological bias of capital or labor [16]. The long-term investment and development of a regional bias will lead to technological progress such as independent research and development in the region [17]. Based on the research of Acemoglu and Lerner, this study takes the derivative of Eqs (1) and (2), and substitutes the corresponding factor selection for each development region, so that the marginal output of capital and labor can be obtained, namely [16, 18]:

$$MP_{L'} = (1-\gamma)N'_L{}^{\frac{\kappa'-1}{\kappa'}}L'^{-\frac{1}{\kappa'}}P_Y{}^{\frac{1}{\kappa'}} \tag{14}$$

$$MP_{K'} = \gamma N'_K{}^{\frac{\kappa'-1}{\kappa'}}K'^{-\frac{1}{\kappa'}}P_Y{}^{\frac{1}{\kappa'}} \tag{15}$$

$$MP_L = (1-\gamma)N_L{}^{\frac{\kappa-1}{\kappa}}L^{-\frac{1}{\kappa}}P_Y{}^{\frac{1}{\kappa}} \tag{16}$$

$$MP_K = \gamma N_K{}^{\frac{\kappa-1}{\kappa}}K^{-\frac{1}{\kappa}}P_Y{}^{\frac{1}{\kappa}} \tag{17}$$

And the technological selectivity bias of independent research and development in different economies are:

$$G' = \partial\left(\frac{MP_{K'}}{MP_{L'}}\right)\Big/\partial\left(\frac{N'_K}{N'_L}\right) = \frac{\kappa'-1}{\kappa'}\frac{\gamma}{1-\gamma}\left(\frac{K'}{L'}\right)^{-\frac{1}{\kappa'}}\left(\frac{N'_K}{N'_L}\right)^{-\frac{1}{\kappa'}} \tag{18}$$

$$G = \partial\left(\frac{MP_K}{MP_L}\right)\Big/\partial\left(\frac{N_K}{N_L}\right) = \frac{\kappa-1}{\kappa}\frac{\gamma}{1-\gamma}\left(\frac{K}{L}\right)^{-\frac{1}{\kappa}}\left(\frac{N_K}{N_L}\right)^{-\frac{1}{\kappa}} \tag{19}$$

In the above formula, $MP_{L'}$, $MP_{K'}$, $G'$ represent the variables of relatively backward economies, while $MP_L$, $NP_K$, $G$ represent the relative variables of relatively developed economies. By importing Eq (13) into Eq (18), we can get:

$$G' = \frac{\kappa'-1}{\kappa'}\left(\frac{\gamma}{1-\gamma}\right)^{\frac{\kappa'\alpha}{1+\kappa'\alpha}}\left(\frac{K'}{L'}\right)^{\frac{\kappa'-1}{\kappa'\delta+\kappa'^2\delta\alpha}}\left(\frac{\delta_K}{\delta_L}\right)^{-\frac{1}{1+\kappa'\alpha}}\left(\frac{N'_K}{N'_L}\right)^{-\frac{\alpha}{1+\kappa'\alpha}}\left[\frac{f'[\eta(T_K)]}{f'[\eta(T_L)]}\cdot\frac{\eta'(T_K)}{\eta'(T_L)}\right]^{-\frac{1}{\kappa'\alpha}} \tag{20}$$

Finally, after transforming Eq (19) into Eq (20), we can get:

$$G' = \left(\frac{\kappa-1}{\kappa}\right)^{\frac{-\kappa\alpha}{1+\kappa\alpha}}\frac{\kappa'-1}{\kappa'}\left\{\frac{\delta_L L}{\delta_K K}\left[\left(\frac{\gamma}{1-\gamma}\right)^{-\kappa}\left(\frac{\gamma'}{1-\gamma'}\right)^{\kappa'}\left(\frac{KL'}{LK'}\right)\right]^{\alpha}\right\}^{\frac{1}{1+\kappa'\alpha}}\left[\frac{f'[\eta(T_K)]}{f'[\eta(T_L)]}\cdot\frac{\eta(T_K)}{\eta(T_L)}\right]^{-\frac{1}{\kappa'\alpha}}(G)^{\frac{\kappa\alpha}{1+\kappa'\alpha}} \tag{21}$$

From Eq (21) above, it can be seen that fiscal tax and fee reduction policies ($\eta(T')$) can affect the technological bias of independent research and development in a region. At the same time, the more developed economies ($G$) have technology spillover to the underdeveloped regions ($G'$). The specific action mode is determined by the factor substitution elasticity of the two economies ($\kappa, \kappa'$) and the related technology spillover intensity coefficient ($\alpha$). Based on this, research hypothesis H1 of this study can be put forward:

H1: Fiscal tax and fee reduction policies are closely related to the technological preference of the region, and act on the surrounding areas by spillover effect, which have certain externality.

Further, the Eq (21) can be transformed into:

$$\left(\frac{\kappa-1}{\kappa}\right)^{\frac{\kappa\alpha}{1+\kappa\alpha}} = \frac{\kappa'-1}{\kappa'}\left\{\frac{\delta_L L}{\delta_K K}\left[\left(\frac{\gamma}{1-\gamma}\right)^{-\kappa}\left(\frac{\gamma'}{1-\gamma'}\right)^{\kappa'}\left(\frac{KL'}{LK'}\right)\right]^{\alpha}\right\}^{\frac{1}{1+\kappa'\alpha}}\left[\frac{f'[\eta(T_K)]}{f'[\eta(T_L)]}\cdot\frac{\eta(T_K)}{\eta(T_L)}\right]^{-\frac{1}{\kappa'\alpha}}\frac{G^{\frac{\kappa\alpha}{1+\kappa'\alpha}}}{G'} \tag{22}$$

From Eq (22) above, it can be seen that local fiscal competitive expenditure strategy will act on the selection of labor and capital factors, specifically affecting the elasticity of substitution of labor and capital factors ($\kappa, \kappa'$). At the same time, focusing on Eq (22), the coefficient $\kappa$ on the right of the equal sign changes with the variation of the coefficient coefficient on the left, which reflects the imitation and assimilation of the production types of factors in developed economies in underdeveloped regions. Along with the selective agglomeration of factors in the two economies, the spillover expansion of any of the advantages of the adjacent areas will occupy the choice of the advantages of the independent research and development of the surrounding areas. Based on this, hypothesis H2 can be put forward:

H2: Tax and fee reduction policies will affect the selection of labor and capital factors for independent research and development in this region. At the same time, this advantageous factor will lead the technological research and development direction of the whole region by being imitated by neighboring regions.

However, combining with the analysis hypothesis H1, H2 and the corresponding assumptions, it reveals the existence of another case that regional tax and fee reduction policies can directly affect the choice of local independent research and development technology elements, and the advantage of technological progress factor will be closely related to the adjacent area of fiscal policy. This reveals that tax policy overflow is not a one-way diffusion, but is a bidirectional regulation effect. Based on the above analysis, hypothesis H3 can be put forward:

H3: The choice of tax and fee reduction policies on the advantageous elements of independent research and development is not a single spillover. The externality of tax and fee reductions on independent R&D is not one-way, but has a two-way effect of spillover and feedback. The developed regions not only serve as the starting point of technology spillover, but also receive feedback from the underdeveloped regions.

## 2.2 Spatial Durbin Model (SDM) method

Tax and fee reduction policies have a significant impact on independent innovation, and the scope of impact is not limited to fixed geographical space or administrative divisions. Due to the externality of economy, the specific ways of policy effect and innovation are also different. Is it the expression of direct effect or indirect effect? Is there regional feedback? In order to verify the externalities and spatial spillovers of tax and fee reduction policies, and examine their

impact on independent research and development, this study uses the Spatial Durbin Model (SDM) proposed by Lesage and Pace for spatial model fitting [19].

There are three general spatial models: Spatial Lag Model, Spatial Error Model and Spatial Durbin Model [20]. We need pay more attention to the parameters in the three model, because the transformation of parameters will define the model. The traditional effect estimation model generally selects Panel Data Model, Differences-in-Differences, Regression Discontinuity and other policy evaluation methods, but the traditional methods cannot achieve the spatial correlation between variables. The traditional measurement model is believed that individuals are independent of each other, but in real life, we should recognize that individuals are interconnected and communicate with each other [21]. For example, the coastal provinces are economically developed and geographically close, which shows the spatial connection; Due to the existence of geographical distance, there will be a certain relationship between individuals, which is represented by the spatial weight matrix. The spatial model, we use different weight matrices such as adjacency and distance to define the geographical relationship between individuals. Different matrix calculation methods are different, and the matrices can be nested. The explained variables are not only influenced by spatial distance, but also by the degree of correlation between economic activities. The model sets as follows:

$$Y_{it} = \alpha + \rho \sum \omega_{ij} Y_{jt} + \beta E_{it} + \gamma \sum \omega_{ij} E_{jt} + \delta C_{it} + \mu_i + \lambda_i + \varepsilon_{it} \qquad (23)$$

Since the 1970s, the rapid development of geographic information technology has promoted the increasingly rich spatial data. The development of spatial data has caused many scholars to pay attention to and investigate the spatial location factors in the field of regional development. The spatial effect has led to great doubts about the independence assumption of variables and the reliability of regression parameters in traditional econometrics, and spatial econometrics came into being. Anselin put forward the classic definition of spatial econometrics, and put the spatial interaction and spatial structure of economic activities into the consideration of econometrics, and established a spatial model. He believed that the spatial effect of the spatial relationship between the variables could be divided into spatial correlation and spatial heterogeneity. The introduction of spatial effect is the main difference between spatial econometrics and traditional econometrics. Spatial correlation refers to the mutual influence of various variables in different locations, that is, the observed values in different spatial locations are consistent, such as spillover and proximity effects, which lead to the influence of a variable on other variables around. Spatial Durbin Model (SDM) is a combination and extension of Spatial Lag Model and Spatial Error Model, which can be established by adding corresponding constraints to the Spatial Lag Model and Spatial Error Model. It is an enhanced SAR model (Spatial Lag Model) by adding spatial lag variables. The SDM is characterized by taking into account the spatial lag correlation between the explanatory variable and the explained variable. In order to analyze all the effects of the explanatory variables in detail, they are divided into direct effects and indirect effects according to their sources. Among them, the direct effect can be divided into two types, one is the direct influence of the explanatory variable on the explained variable, and the other is the feedback effect caused by the independent variable influencing the dependent variable in the adjacent region; The indirect effect is the spatial spillover effect of explanatory variables, which can also be divided into two types. One is the influence of the independent variables of neighboring regions on the dependent variables of the region, and the other is the influence of the independent variables of neighboring regions on the dependent variables of the region.

The above equation is the concrete construction form of static Spatial Durbin Model, and on the basis of testing whether the above equation has significant direct and indirect effects, the dynamic Spatial Durbin Model can be further constructed. On the one hand, in order to ensure the comprehensiveness of spatial panel data information. On the other hand, partial endogenity can be overcome to some extent by introducing the lag period of explained variables. The specific setting of dynamic Spatial Durbin Model is as follows:

$$Y_{it} = \alpha + \sigma Y_{it-1} + \rho \sum \omega_{ij} Y_{jt} + \beta E_{it} + \gamma \sum \omega_{ij} E_{jt} + \delta C_{it} + \mu_i + \lambda_i + \varepsilon_{it} \qquad (24)$$

Eq (24) explains the acting conditions and spillover externalities of the effect of tax and fee reduction policies on independent research and development, and illustrates the dynamic spatial effect of the connection between the effect of tax and fee reduction policies ($E_{it}$) and independent research and development ($Y_{it}$). Where, $i$ and $t$ are the parameters of cities and years selected for analysis in this study. $\omega_{it}$ is the spatial weight matrix set in this chapter and $C_{it}$ is other control variables. $\mu_i$ and $\lambda_i$ represent the fixed effect of city and time respectively, and $\varepsilon_{it}$ is a random error term. $Y_{t-1}$ is the explained variable lag, and the dynamic coefficient of $\rho$ is measured to independent research and development level of spatial autocorrelation coefficient. $\beta$ represents the direct effect coefficient of tax and fee reduction policies, and $\gamma$ is the spatial spillover effect coefficient after the implementation of the policy. And the parameters of each control variable are to be set as $\delta$. Then the above parameters are the parameters to be estimated in this study.

## 2.3 Variables and data sources

**2.3.1 Selection of variables.** *(1) Explanatory variables*. Tax and fee reduction variables: Most literature focuses on changes in tax burden rates and the reduction of fees [22]. However, China has a wide range of taxes and diverse charging methods, and there is currently no effective way to account for all changes. Considering that fiscal revenue includes tax revenue and non tax revenue, we start from the concept of small finance and choose the change in fiscal revenue as the variable of tax and fee reduction, while retaining the corresponding symbols of the data to cope with negative changes. In order to mitigate data fluctuations and control the heteroscedasticity, the variable is logarithmized, and the difference form of the indicator in the previous and last two years is used.

It should be emphasized that the policy effect of reducing taxes and fees is often directly reflected through fiscal or tax competition among local governments [23, 24]. Most studies choose the proportion of public financial expenditure to measure the intensity of local competition [25], but economic research data are limited by personnel, methods and tools, and often have deviations from the actual information. This study uses DMSP/OLS night light data to describe local financial Porter's generic strategies, to replace the main explanatory variables for robustness testing [26]. Night light data, as a description index of regional economy such as infrastructure and economic development level, can overcome the false information in economic research data and avoid the endogeneity of reverse causality between economic variable data.

DMSP/OLS Night Lights Raw Data: DMSP originated from the U.S. Department of Defense's Weather Satellite Program. The data selected in the paper are the non-radiometric calibrated version IV published by NGDC. We use the fixed target area method as the correction method for acquiring light images, and use the Lambert equiangular cone form for re projection of DMSP/OLS night light data. We also adjust the resampling range to 1 kilometer, and then crop it to the average intensity of a non-radiometric calibration form with a spatial resolution of 1 kilometer for extraction and denoising. The time span of the nighttime light data

we can collect is 2000–2013, involving 243 Prefecture-level city nationwide. Finally, the composite average value of the light intensity collected by each province is taken as the substitute variable for the robustness test.

*(2)Control variables.* A series of other control variables are mainly to measure the level of local economic development, because the level of economic development in a region usually determines the endogenous driving force of independent research and development [27]. The specific indicators selected in this study are: economic growth rate, population dependency ratio, population growth rate, opening level, consumer price index, unemployment rate. Among them, the economic growth rate has a strong correlation with the level of independent research and development of a country or region; The degree of urbanization can represent the overall development level of a city, and can also be directly used to evaluate the economic development level and its derivative can partly represent the growth rate; Talents are regarded as the basis of innovation, so population variables, such as dependency ratio and Population growth, are introduced in this paper; As a description of the development level of an economy, indicators such as openness level, consumer price index and unemployment rate all reflect the realization basis of independent research and development, which also need to be introduced; The degree of openness of a region to the outside world determines the impact of foreign investment on the economy, and we have chosen foreign direct investment/per capita GDP to represent it. Detailed variables and data are shown in Table 1. And to control the heteroscedasticity, the logarithm of the above variables is taken.

*(3) Explained variables.* The ratio of R&D expenditure to operating income is used to interpret the level of independent research and development. Furthermore, this article replaced the dependent variable in the robustness test. Decompose the Malmquist productivity index to obtain an indicator that characterizes the level of independent innovation— the technological progress index. The technological progress index can reflect R&D investment, output capacity, and the independent innovation situation of enterprises. Malquist index is a measure of the output input ratio, used to assess production efficiency and can be decomposed into several sub efficiency indicators. The calculation method of Malmquist index can be expressed by a formula: Malmquist index can be decomposed into technical progress index and technical efficiency change index. Among them, the technological progress index is the ratio of productivity indices at two time points, and the technological efficiency change index is the ratio of productivity indices at two time points divided by the technological progress index. The Malmquist technology progress index is selected in this

**Table 1. Variable description and statistical characteristics.**

| variable name | mean | standard deviation | minimum value | Maximum value |
|---|---|---|---|---|
| Independent research and development level | 4.93 | 4.60 | 0.44 | 6.52 |
| policy effect | 4.71 | 4.96 | -1.02 | 6.84 |
| consumer price index | 4.77 | 0.20 | 4.52 | 5.47 |
| degree of urbanization | -0.84 | 0.36 | -1.96 | -0.11 |
| dependency ratio | 3.64 | 0.21 | 2.96 | 4.17 |
| population growth rate | 1.51 | 0.76 | -1.47 | 2.77 |
| GDP per capita | 9.78 | 0.85 | 7.77 | 11.59 |
| unemployment rate | 1.23 | 0.26 | -0.51 | 1.87 |
| foreign direct investment | 4.87 | 1.92 | -0.11 | 9.30 |
| Malmquist technology progress index | 1.59 | 0.66 | 0.96 | 2.03 |

manuscript, and the specific calculation is: $T^t = \frac{D_N^t(x^{t+1}, y^{t+1})}{D_N^t(x^t, y^t)}$. $x$ represents the R&D expenditure for period $t$, which is the indicator value of scientific and technological progress investment. $y$ represents the output profit value of period $t$, which is the indicator value of scientific and technological progress output, and $D_N^t$ is the distance function. See Table 1 for detailed data description.

**2.3.2 Data sources.** The spatial model established in this study intends to construct estimation at the provincial level. Among them, the main data of the sample of 31 provinces came from *China Statistical Yearbook*, *China Financial Yearbook*, *China City Statistical Yearbook*, *China Statistical Yearbook*, *High-tech Industry Statistical Yearbook* and the websites of the National Bureau of Statistics and the Ministry of Finance from 1998 to 2019. Restricted by the availability and integrity of the collected data, Hong Kong, Macao and Taiwan have a lot of missing data, so this study eliminated them. At the same time, due to the intertemporal behavior of tax and fee reduction policies and independent research and development, we list the spatial lag items of both, so as to measure the policy impact of reducing taxes and fees. In addition, the economic structure and population composition of Tibet are relatively special. The urbanization degree from 1998 to 2004 and the unemployment rate from 1998 to 2008 are missing except for 2000, 2002 and 2004. In this study, Tibet is also deleted and only listed as the control group for comparison and verification. And Shanghai's unemployment figures for 2001 and 2005 were also missing. When making model estimation, it can be automatically ignored by Stata, which is not dealt with in this study for the time being. To sum up, the final sample is the data of provincial administrative regions except Hong Kong, Macao and Taiwan with a time span of 22 years.

From the setting and estimation process of the spatial model in this study, it is necessary to set the spatial weight matrix first. Among them, spatial economics considers that geographical attributes are related to the same attributes of adjacent spatial units. On this basis, most literatures introduce neighboring weight matrix to represent geographical features. However, when capturing prefecture-level data in this chapter, most cities between regions are not in the form of car adjacent or rear adjacent, but are clustered in concentrated areas of provinces, and have relatively similar geographical and economic attributes. Therefore, according to the geographical factors studied, we does not select the neighboring weight matrix, but constructs the geographical distance weight matrix. In addition, the theorem of geography is put forward that the closer the distance between things is, the closer the connection will be. That is, the correlation between regions and the distance between them presents a negative correlation. According to the above analysis, the Euclidean distance index is selected in this study to set the weight matrix of geographical distance:

$$\omega_{ij} = \begin{cases} {}^1/_{d^2} & i \neq j \\ 0 & i = j \end{cases} \tag{25}$$

The above formula represents the Euclidean distance between two intervals. In order to improve the tolerance of weight matrix setting for economic differences, we introduce the weight matrix of economic distance based on the established geographical distance matrix. In addition, the research on tax and fee reductions involves the field of public finance, so we have to consider the background of fiscal decentralization. Under the fiscal decentralization system, local governments conduct scale competition, which leads to obvious diffusion and spillover effect between different regions [28]. Therefore, in addition to introducing the weight matrix of economic distance with GDP difference as the main feature, we construct another weight

matrix of fiscal decentralization distance:

$$\omega_{ij} = \frac{1}{d^2} \cdot \frac{1}{|PGDP_i - PGDP_j|} \qquad (economic\ distance\ matrix) \qquad (26)$$

$$\omega_{ij} = \frac{1}{d^2} \cdot \frac{1}{|FDZ_i - FDZ_j|} \qquad (fiscal\ decentralization\ distance\ matrix) \qquad (27)$$

$PGDP_i$ and $PGDP_j$ represent the actual per capita GDP levels of the two regions in 2001 (the base period is 2000), while $FDZ_i$ and $FDZ_j$ represent the degree of fiscal decentralization of the two regions. Considering from the samples of prefecture-level city level data, the ratio of local income to local expenditure is used to measure in this study. The setting of the above city weight matrix was captured by ArcGIS using the 2000 national large coordinate system, and obtained after forming a sparse matrix by MATLAB. Some data changes were added or decreased by pure manual method.

## 3. Results and discussion

### 3.1 Spatial autocorrelation test

Spatial autocorrelation statistics are a fundamental property used to measure geographic data: the degree of interdependence between data at a certain location and data at other locations. Usually, this dependency is called spatial dependency. Due to the influence of spatial interaction and diffusion, geographic data may no longer be independent of each other, but rather related. Spatial autocorrelation is an important starting point for the setting of spatial models. It is a method to calculate the correlation of a variable to a spatial location and an effective way to measure the aggregation degree of adjacent areas in space. The spatial autocorrelation test can be divided into global and local spatial autocorrelation according to the size of the set spatial range. At present, in the field of spatial research, there are many methods for testing and estimating spatial autocorrelation. However, the mainstream economic works and research literatures show that most scholars generally agree on the estimation method of Moran's I index, which can make the most effective statistics on spatial distribution characteristics and differences. Moran's I index covers not only global Moran's I index, but also local Moran's I index, which is also known as LISA. We use the Moran's I index for spatial autocorrelation test, where the spatial weight matrix selects the spatial adjacency matrix (0–1 matrix). The spatial weight of adjacent provinces is 1, and the spatial weight of non adjacent provinces is 0. The Moran's I index is an indicator used to measure spatial correlation, and its core is the spatial adjacency matrix. Before setting the spatial model, we first tested the spatial correlation and specifically calculates the global Moran's I index for the level of independent research and development, which was estimated by GEODA in this study.

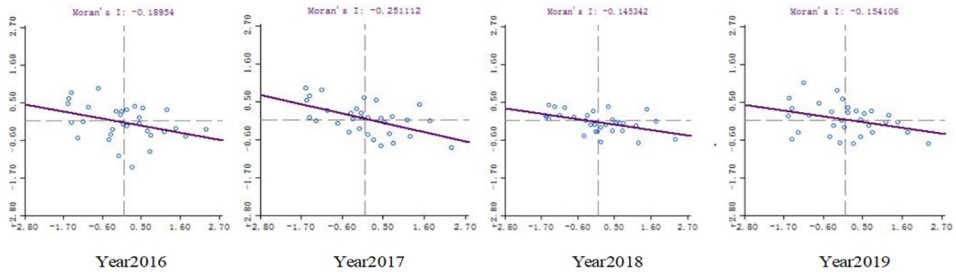

**Fig 1. Moran's I index from 1996 to 2019.**

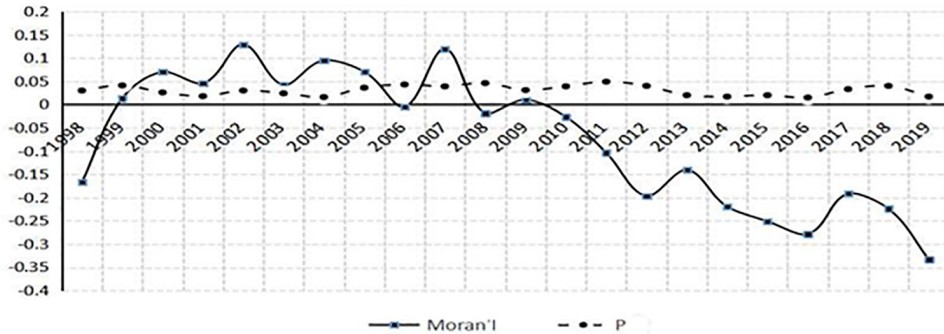

**Fig 2. Moran's I index and corresponding P value from 1998 to 2019.**

Fig 2 shows the line chart of Moran's I index and corresponding P value in each year from 1998 to 2019. It can be seen from the data in the figure that the value of Moran's I index can be positive or negative, and the corresponding P value is within the significance level of 5%, which indicates that there is spatial correlation between independent research and development. The positive or negative spatial autocorrelation needs to be further verified. In addition, on the basis of measuring local Moran's I index, we also draw a scatter plot of Moran's I index, as shown in Fig 1 in each year from 1996 to 2019 (due to space limitations, please refer to the attached Moran's I Index chart for the year 1998–2019). The Fig 1 shows that the spatial similarity of independent research and development of various provinces in China is obvious in a certain period, while the spatial heterogeneity in a certain period also indicates that there is no consistency in the spatial dependence characteristics, and the matching types of neighboring regions have different attributes. Figs 1 and 2 show that the model is spatially dependent, and the Moran's I exponent symbol is not uniform,which indicate that the spatial autocorrelation may be positive, and there is a negative possibility.

In order to test the spatial characteristics and determine the rationality of the Spatial Lag Model and Spatial Error Model, we also add the robust LM test to demonstrate the adaptability of the spatial model. As shown in Table 2, the adjacent weight matrix is selected for estimation in the study, which respectively corresponds to the joint OLS estimation, spatial fixed effect,

**Table 2. No interaction effect estimation results and spatial diagnostic test.**

| variable | (1) | (2) | (3) | (4) |
|---|---|---|---|---|
| policy effect | -24.63* (-1.76) | -32.60** (-2.36) | -1.66*** (-7.28) | -8.42*** (-38.27) |
| consumer price index | 1.16*** (5.00) | 1.25*** (5.48) | 0.45** (1.98) | 0.57** (2.53) |
| dependency ratio | -1.37 (-1.64) | -1.08 (-1.30) | -3.60*** (-3.90) | -3.27*** (-3.53) |
| other control variables | YES | YES | YES | YES |
| sigma^2 | 9203.15 | 8748.79 | 6465.11 | 1493.756 |
| Log-L | -3333.75 | -3320.13 | -3235.73 | -3220.79 |
| LM-lag | 345.69*** | 345.45*** | 29.72*** | 25.55*** |
| Robust LM-lag | 0.04* | 0.96* | 1.38* | 1.77* |
| LM-error | 364.63*** | 360.19** | 36.70*** | 32.38*** |
| Robust LM-error | 18.98*** | 15.70*** | 8.35*** | 8.60*** |

Note: The parentheses in the table are T values.

***, ** and * represent 1%, 5% and 10% significance levels.

time fixed effect and bidirectional fixed effect models. The specific estimated parameters are detailed in the Table 2.

As shown in Table 2 above, both the traditional LM estimate and the robust LM test estimate reject the null hypothesis. This indicates that the spatial model should contain the spatial lag explained variable and the spatial error term. The specific details are as follows: when setting up the traditional LM test, the model estimates significantly reject the original hypothesis without spatial lag terms (the dependent variable) and spatial error terms. However, when using the robust LM estimation test, which states that the robust LM test with both time and space fixed effects do not pass the hypothesis test, all other types reject the null hypothesis without spatial lag dependent variables except for Eq (4). This can be shown that the spatial delay explained variable and the spatial error term should be set in the estimation of the spatial model. In addition, the likelihood ratio LR test is used to test the non-significant joint null hypothesis of spatial fixed effect. The results were as follows: the null hypothesis of joint non-significance of spatial fixed effect was rejected (the estimated value of this test: 429.89, degrees of freedom: 31, $P < 0.01$); The null hypothesis of the combination of time fixed effect and non-significance was also rejected (the estimated value of this test: 198.68, degrees of freedom: 18, $P < 0.01$). According to the above results, the spatial model should adopt the bidirectional fixed effect model, that is that the spatial model should be extended into the form of bidirectional fixed effect model with space fixed effect and time fixed effect.

After selecting the bidirectional fixed effect model as the basic form of the spatial model, the Spatial Durbin Model should be constructed initially, and then the Spatial Lag Model or Spatial Error Model should be selected through relevant hypothesis testing. Among them, if all the null assumptions are rejected after estimating the parameters of the model, then we should adopt the Spatial Durbin Model for subsequent studies. On the contrary, we should choose whether to build Spatial Lag Model or Spatial Error Model through specific test indicators. The above two kinds of tests are both subject to chi-square distribution, and LR test or Wald test is usually used in the form of test. The advantages and disadvantages of the two kinds of tests are obvious. In order to ensure the accuracy and integrity of the test, the calculation and estimation of the two kinds of tests are carried out in this study. The estimation results are

**Table 3. Static space model checking.**

| variable | space fixed effect period fixed effect | bidirectional fixed effect bias correction | spatial random effect period fixed effect |
|---|---|---|---|
| spatial lag for independent research and development | 0.49*** (6.81) | 0.66*** (12.93) | 0.50*** (7.15) |
| policy effect | 2.77*** (2.96) | 2.79*** (2.88) | 3.00*** (3.18) |
| spatial lag for policy effect | -90.87** (-2.18) | -87.39**(-2.03) | -90.08** (-2.13) |
| other control variables | YES | YES | YES |
| teta | | | 0.99*** |
| sigma^2 | 5436.23 | 5855.39 | 5759.12 |
| Log-likehood | -3196.55 | -3196.55 | -3212.77 |
| Wald spatial lag | 24.62*** | 24.76*** | 18.80** |
| LR spatial lag | 24.12*** | 24.12*** | 18.60** |
| Wald spatial error | 18.06** | 15.31* | 12.75* |
| LR spatial error | 17.61** | 17.61** | 8.27* |

Note: The parentheses in the table are T values.

***, ** and * represent 1%, 5% and 10% significance levels.

shown in Table 3. Since the static space panel is set in the first step, we further construct ML estimation for the static space panel, and test its spatial fixed effect, time fixed effect and bidirectional fixed effect respectively, which correspond to columns (1)—(3) in Table 3 below. Among them, the Wald and LR estimates of the three static spatial models all passed the significance test, and all refused to be transformed into Spatial Lag or Spatial Error Model. In addition, the estimated value of teta, which is the difference parameter between the fixed-effect model and the random-effect model is 0.99 and very significant. Combined with Hausman's estimation results (estimated value: 33.687, degree of freedom: 13, p = 0.0013<0.01), it is shown that the spatial model should not exclude fixed effect. To sum up the estimation results, the Spatial Durbin Model of bidirectional fixed effect is used for further analysis in this study.

From Table 3 above, it can be seen that the spatial model selected in this study can be further extended into a static Spatial Durbin Model with double fixed effect. In addition to testing the spatial properties of the model constructed in this study, the estimated value in Table 3 above shows that the spatial lag term of independent research and development of the explained variable is positive, which indicates that the independent research and development of local provinces can affect the independent research and development level of neighboring provinces with certain positive externalities. At the same time, it focuses on the policy effect of reducing taxes and fees. The local province is affected by the effect of local tax and fee reduction policies, which is reflected in the improvement of independent research and development level, while the neighboring provinces are affected by negative externalities, which restrains their independent research and development process. It is worth emphasizing that the specific estimated parameters of the static Spatial Durbin Model do not represent the marginal effect. Focusing on the analysis of time lag in the following study, the effect can be decomposed into short-term and long-term coefficients. In this study, the tax and fee reduction policies are estimated to have positive externalities and the policy shows positive income effect in the local area and negative substitution effect in the neighboring area. This makes the regional independent research and development show a point with the clustering, which is gradually form a central point of the agglomeration phenomenon.

## 3.2 Dynamic Spatial Durbin model estimation

The spatial lag items of the main variables listed in Table 3 above have basically passed the significance test, which indicates that the policy effect of local finance and the process of independent research and development present a strong spatial dependence. The effect of tax and fee

**Table 4. Dynamic Durbin model estimation of spatial interaction.**

| variable | adjacency matrix | geographic distance matrix | economic distance matrix | fiscal decentralization matrix |
|---|---|---|---|---|
| policy effect | 3.63*** (3.60) | 3.19*** (2.94) | 3.56*** (3.51) | 3.47*** (3.45) |
| W*policy effect | -6.82** (-2.02) | -0.63*** (-2.85) | -3.86** (-2.49) | -1.69*** (-2.80) |
| time lag items of independent research and development | -3.63*** (-3.60) | 0.12** (1.82) | -0.60** (-2.49) | -0.64*** (-2.63) |
| W*independent research and development level | 0.99*** (7.04) | 0.21*** (2.95) | 0.08** (2.26) | 0.97*** (3.07) |
| W*time lag of independent research and development | 0.41* (1.79) | -0.15* (-1.80) | 3.66* (1.93) | -1.22** (-1.96) |
| other control variables | YES | YES | YES | YES |
| Log-likelihood | -3022.15 | -3040.05 | -3041.06 | -3038.78 |

Note: The parentheses in the table are T values.

***, ** and * represent 1%, 5% and 10% significance levels.

reduction policies and the level of independent research and development of neighboring provinces will affect that of their own provinces. In order to effectively control the endogeneity of the time trend item and to more carefully describe the impact of the policy effect of reducing taxes and fees on the level of independent research and development, this chapter chooses the dynamic Spatial Durbin Model for further analysis, and its specific estimates are shown in Table 4 below. As it can be seen, the estimated value of LR is 2× (-3022.15–3196.55) = -12437.4, the degree of freedom is 2, p<0.01, which indicates that the static space panel can be extended to the dynamic space panel model for estimation. The estimated parameters of the specific dynamic Spatial Durbin Model are shown in Table 4 below.

As shown in Table 4, various parameter estimates of the dynamic Spatial Durbin Model are estimated in the above table. Among them, by comparing the Spatial Static Durbin model, it can be found that after the introduction of the time lag term and the explained variable of the space lag term, the sign of the variable coefficient of the effect of tax and fee reduction policies has been greatly shifted. This indicates that the tax and fee reduction policies of this province have an externality effect, which presents regional restraining effect on the independent research and development process of neighboring provinces, while the local fiscal policy strategies only aim to improve the independent research and development level of this region. Furthermore, whether space and time lag items are added or not, the coefficient of independent research and development level is basically positive, which indicates that the specific effect of local independent research and development level is easy to form a regional pattern, which can accelerate the process of regional independent innovation from point to area. At the same time, after introducing the spatial weight matrix of geographical factors, economic factors and political decentralization factors into the dynamic Spatial Durbin Model, the policy effect of reducing taxes and fees and the process of independent research and develpoment do not show the sign fluctuation, which further guarantees the accuracy of parameter estimation in this chapter.

## 3.3 Partial differential decomposition of dynamic space effects

In the model estimation of spatial metrology, the direct coefficient interpretation of dynamic Spatial Durbin Model does not represent the marginal effect. The estimated value of each parameter in Table 4 above can not be used to describe the marginal change quantity. Therefore, partial differential decomposition of the coefficient estimates above is performed in this section to obtain the specific action form of policy effects. The results are shown in Table 5 below.

As the Table 5 estimates, regardless of what kind of spatial weight matrix, the building such as adjacency matrix, geographic distance matrix and economic distance matrix, fiscal

**Table 5. Dynamic spatial effect decomposition.**

| variable form | effect decomposition | adjacency matrix | geographic distance matrix | economic distance matrix | fiscal decentralization matrix |
|---|---|---|---|---|---|
| effect in the short | direct effect | 6.42*** (27.61) | 15.66*** (12.94) | 17.97*** (11.09) | 14.73*** (10.91) |
| | indirect effect | -75.11*** (-82.99) | -7.77*** (-6.52) | -5.31** (-2.43) | -4.79*** (-4.62) |
| effect for a long | direct effect | -35.24*** (-27.88) | -15.64*** (-10.93) | -17.85*** (-10.09) | -14.27*** (-10.89) |
| | indirect effect | -947.90*** (-1011.23) | -4.57** (-2.28) | -6.22** (-2.17) | -1.25** (-2.34) |

Note: The parentheses in the table are T values.

***, ** and * represent 1%, 5% and 10% significance levels. Since the decomposition of the correlation coefficients of other control variables are not included in the main research content of this study, they are not listed here. Please ask for the author of this study if necessary.

decentralization matrix (corresponding to the Table 5 column (1)—(4)), those do not affect the dynamic space doberman model of partial differential coefficient decomposition, and the results are significant. In the short term, the direct effect is positive, while the indirect effect is negative. However, in the long run, no matter the direct effect or the indirect effect, no matter the long-term effect or the short-term effect, the estimated results of the effect of tax and fee reduction policies are negative. That is to say, the tax and fee reduction policies of this province can not only act on the local independent research and development, but also affect this of neighboring provinces through external spillover.

In the short term, no matter the adjacency matrix, geographical distance matrix, economic distance matrix or fiscal decentralization matrix, the policy effect of reducing taxes and fees significantly promotes local independent research and development, which is in line with the essence of national innovation development strategy. However, when focusing on indirect effects, no matter the adjacency matrix, geographic distance matrix, economic distance matrix, or fiscal decentralization matrix, the policy effect of reducing taxes and fees is shown as the negative externality of neighboring provinces, which tends to inhibit independent research and development process of neighboring provinces. That is to say, from the perspective of space, the positive incentive of local policy effect on enterprises' independent research and development is only limited within the administrative planning area. Neighbors may be affected by local financial competition or economic development level differences, and can only rely on regions with strong independent research and development, thus forming the agglomeration phenomenon of a center and affiliated surrounding areas. The surrounding regions are highly dependent on each other, and they are tired of independent research and development in a short term. This phenomenon often proves that the neighboring provinces lag behind in certain innovation. They only adopt the introduction, dependence and reintroduction method through domestic purchase, and further lose the core competitiveness of independent innovation.

In the long run, no matter the adjacency matrix, geographical distance matrix, economic distance matrix or fiscal decentralization matrix, the policy effect of reducing taxes and fees will inhibit the independent research and development level of both local and neighboring provinces. This phenomenon also once again confirms that local enterprises have different behaviors in response to government incentives for tax and fee reduction policies. There are two possible reasons: on the one hand, local governments may completely abandon incentives for local independent research and development. The reason may be the contradiction between the long-term nature of independent research and development and the short tenure of local governments. In the short term, local governments are prone to pressure from political promotion incentives or economic performance evaluations, and usually implement fiscal and tax policies that stimulate independent research and development [29]. However, the term of office is short (generally 3–5 years), and the departure of officials means the separation of the relationship with the original administrative region.

As a result, local governments only pay attention to short-term interests and do not take long-term considerations when implementing the incentive policies of reducing taxes and fees. On the other hand, local enterprises may have strategic behaviors in response to tax and fee reduction policies, which confirms the research in the previous chapter of this study. That is, enterprises may choose different disposal methods based on different motivations for independent research and development. This creates the marginal effect of long term policy to drop. On the basis of measuring the material cost and time cost of technology introduction and independent research, enterprises are easy to choose to give up independent research and development. The related neighboring provinces, due to the lag of the overall regional technology level, also adopt the innovation path of foreign technology introduction→ dependence → reintroduction. This leads to the process of regional independent innovation to appear

inhibitive. Meanwhile, it also further verifies the research hypothesis 3 of this study: when enterprises respond to tax and fee reduction policies, they will be motivated by different purposes and present strategic innovation behaviors, which will affect the choice of innovation path of enterprises.

It should be noted that no matter which matrix the above spatial weight is set to, the correlation between provinces is not limited by geographical, economic and institutional administrative barriers. At the same time, the externalities among provinces are not mainly manifested in the form of one-way spillovers within the region: according to calculation, tax and fee cuts have a two-way feedback effect on independent research and development in neighboring regions, and the size of the feedback effect is -8.42+3.44 = -4.98. Although feedback effect is small, but is a very important indicators. The feedback effects show that the policy spillover effect of neighboring provinces by the province to change outside of independent research and development process, but also by spillover effect to give effect to this province, which is verified in this study, the assumption of six: the assumption that the effect of tax and fee reduction policies of the independent research and development is not a single spillover in one-way, but has a two-way effect of spillover and feedback. The more developed areas not only serve as the starting point of technology spillover, but also receive feedback from the level of independent research and development in the less developed areas.

## 3.4 Robustness test

Considering the endogeneity treatment of spatial econometric models, we have constructed static and dynamic spatial models. The introduction of spatial and temporal parameters, as well as the spatial and time lag terms of the explained variables can overcome some endogeneity to a certain extent, and the decomposition of the coefficients of the dynamic Spatial Durbin Model can more purely determine the marginal utility of the parameters, which has good persuasiveness for the subsequent estimation results of this paper. Furthermore, different spatial weight types were constructed, such as the adjacency weight matrix, geographical weight matrix, economic weight matrix, and fiscal decentralization weight matrix, which can improve the explanatory power of the parameters.

To be explained variable selection of heterogeneity and sampling range selection bias, we intend to expand the government competence—using the prefecture level data and the DMSP/OLS night light data to characterize the local finance competition, and explain the local fiscal policy intervention to innovation, in particular, tax and fee reduction policies. In addition, explained variables are replaced in this section. Malmquist productivity index is used to decompose the index representing the level of independent innovation—technological progress index. Technological progress index can reflect the R&D input and output capacity as well as the independent innovation of enterprises [30, 31].

In order to ensure the accuracy of sample selection, we intercept data of the periods, integrate the data structure of prefecture-level cities, integrate the time range that can be captured by DMSP/OLS night light data, and the relevant variables that can be obtained by each prefecture-level city, eliminate the indicators that are seriously missing in the data, and perform logarithmic processing on the data with large fluctuations. Data samples in this section are selected from 243 panel data of prefecture-level cities from 2001 to 2013, with a total of 3159 samples. Furthermore, the dynamic Spatial Durbin Model was used for parameter estimation, and the parameter estimates were similar to those of the model above, and all of them passed the effect decomposition test. The estimation results of Durbin model in dynamic space are shown in Table 6, and the partial differential decomposition results of each parameter effect are shown in Table 7.

**Table 6. Spatial interactive dynamic Durbin model test of night light data.**

| variable | adjacency matrix | geographic distance matrix | economic distance matrix | fiscal decentralization matrix |
|---|---|---|---|---|
| policy effects substitution | 0.32*** (8.58) | 0.399*** (2.844) | 0.34*** (8.92) | 0.34*** (8.89) |
| W*policy effects substitution | -0.01*** (-2.89) | -0.02** (-2.56) | -0.01** (-2.46) | -0.01*** (-3.03) |
| technological progress index time lag item | 0.05* (1.86) | 0.12*** (3.13) | 0.24* (1.65) | 0.01** (2.25) |
| W*technological progress index | -0.01* (-1.85) | -0.01 (-0.02) | 0.01** (1.79) | -0.58* (-1.67) |
| W*technological progress index time lag item | 0.02*** (3.51) | -0.24* (-1.91) | -0.81* (-1.72) | 0.73* (-1.65) |
| other control variables | YES | YES | YES | YES |
| Log-likelihood | -73.07 | -86.75 | -88.16 | -88.39 |

Note: The parentheses in the table are T values.

***, ** and * represent 1%, 5% and 10% significance levels.

**Table 7. Dynamic spatial effect decomposition of night light data.**

| variable form | effect decomposition | adjacency matrix | geographic distance matrix | economic distance matrix | fiscal decentralization matrix |
|---|---|---|---|---|---|
| effect in the short | direct effect | 0.33*** (8.75) | 0.35*** (8.84) | 0.34*** (8.81) | 0.34*** (9.18) |
| | indirect effect | -0.01** (-2.17) | -0.01* (-1.73) | -0.02* (-1.72) | -0.02* (-1.84) |
| effect for a long | direct effect | -0.34*** (-3.47) | -0.36*** (-7.90) | -0.35*** (-7.91) | -0.35*** (-8.23) |
| | indirect effect | -0.11* (-1.75) | -0.08* (-1.66) | -0.04** (-1.99) | -0.01* (-1.92) |

Note: The parentheses in the table are T values.

***, ** and * represent 1%, 5% and 10% significance levels. Since the decomposition of the correlation coefficients of other control variables are not included in the main research content of this study, they are not listed here. Please ask for the author of this study if necessary.

As can be seen from the estimation results in the tables above, after replacing the explained variables and sample range, and introducing the objective indicator DMSP/OLS night light data for variable substitution, the significance of the estimated results is less different from the direction of the policy effect, which basically conforms to the estimation results of the spatial model mentioned above. In conclusion, the direct short-term effect is significant and indirect effect in both the short and long term are characterized by negative significantly, which verified the above estimates—tax and fee reduction policies tend to encourage enterprises to promote the level of local independent research and development in the short term, while its impact on surrounding areas presents a negative externality. In addition, the estimated value of local technological progress index in response to tax and fee reduction policies is significant, which also reflects that there is a time lag in the incentive effect of policy on independent research and development, which is relatively easy to understand, and technological progress has a certain inertia. When an entity attempts to conduct core technology research and development, it often increases research and development expenses due to considerations of cost investment, future returns, and market position [32]. This further accelerates the process of independent research and development.

## 3.5 Further study

The dynamic Spatial Durbin Model was adopted above to empirically test the relationship between the effect of tax and fee reduction policies and the process of independent innovation. However, fiscal policy has the characteristics of wide coverage and long duration, and its influence on independent innovation is complex. Based on the above considerations, this study

**Table 8. Test of spatial interaction between labor and capital factors.**

| variable | labor capital ratio | capital price | labor price | capital productivity | labor productivity |
|---|---|---|---|---|---|
| policy effect | -0.03** (-2.20) | -0.07*** (-2.96) | 1.01* (1.92) | 159.29* (1.66) | 391.42*** (12.31) |
| W*policy effect | -0.04*** (-2.89) | -0.04* (-1.84) | 0.02 (0.14) | 28.15** (2.20) | 226.82* (1.65) |
| $Y_{t-1}$ | -0.07* (-1.68) | -0.13* (-1.92) | 0.21* (1.82) | 30.56** (2.41) | 21.99* (1.93) |
| W*Y | 0.05*** (2.82) | -1.01*** (-9.02) | 0.81** (1.99) | 3.54 (0.10) | -2.25** (-2.22) |
| $W*Y_{t-1}$ | -0.06* (-1.80) | -0.22* (-1.73) | 1.81*** (-3.72) | 0.11* (1.84) | 2.14** (2.47) |
| other control variables | YES | YES | YES | YES | YES |
| Log-likelihood | -99.05 | -188.05 | -408.86 | -2481.46 | -4542.51 |

Note: The parentheses in the table are T values.

***, ** and * represent 1%, 5% and 10% significance levels.

discusses the conduction between policy effect and independent research and development by selecting two main factors of Cochrane's production function that affect technological progress, namely labor and capital. The estimated evidence can provide a reference for the incentive process of independent research and development while further testing the empirical results above. This section continues to use the sample selection criteria of the above studies. On the premise of continuing the existing explanatory variables and control variables, the panel data of 243 prefecture-level cities from 2001 to 2019 are constructed, with a total sample size of 5260. In addition, since the estimated values in this section are mainly economic indicators, the spatial weight matrix of economic distance is introduced and the estimates of other spatial weight matrices are similar. Among them, the specific parameter estimation results are shown in Table 8, and the partial differential of each coefficient is decomposed into Table 9.

According to the estimates in Tables 8 and 9 above, after partial differential decomposition of the partial effects of labor factors and capital factors, it can be seen that all the estimated parameters are basically significant. Tax and fee reduction policies can significantly affect the ratio of labor to capital, their respective price levels and input-output rates. Among them, there is a negative correlation between direct effect and indirect effect on capital price, while there is a positive correlation between labor price, capital input output rate and labor input output rate. This suggests that tax and fee reduction policies can affect the technology selection bias and improve the productivity of two factors by adjusting the price of capital and labor. It is testified the hypothesis four, which is that the tax and fee reduction policies closely associated with area of technology selection bias, and effect on the surrounding areas by spillover. In addition, the effect of the policy of local can affect the surrounding area through the external

**Table 9. Dynamic spatial effect decomposition of labor and capital factors.**

| variable form | effect decomposition | labor capital ratio | capital price | labor price | capital productivity | labor productivity |
|---|---|---|---|---|---|---|
| effect in the short | direct effect | -0.02** (-2.51) | -0.07** (-2.54) | 0.01*** (3.11) | 15.96*** (16.99) | 38.90*** (4.14) |
| | indirect effect | 0.01** (1.97) | -0.03*** (-3.21) | 0.03* (1.95) | 5.42** (2.20) | 24.51** (2.29) |
| effect for a long | direct effect | -0.52*** (-3.62) | -0.07** (-2.52) | 0.32*** (-8.81) | 16.23*** (16.69) | 37.96*** (4.26) |
| | indirect effect | -0.11* (-1.75) | -0.02* (-1.88) | -0.03* (-1.82) | 8.47* (1.68) | 25.71** (2.52) |

Note: The parentheses in the table are T values.

***, ** and * represent 1%, 5% and 10% significance levels. Since the decomposition of the correlation coefficients of other control variables are not included in the main research content of this study, they are not listed here. Please ask for the author of this study if necessary.

overflow and this is reflected in the convergence of labor and capital factors in neighboring areas (both positive and negative prices and productivity). In the long run, the demand of the dominant factors in this region will lead the independent research and development bias of the whole region by means of the factor supply and technology imitation in the neighboring regions. This tests Hypothesis 5 of this study, which is that tax and fee reduction policies will affect the selection of labor and capital factors in the region's preference for independent research and development. At the same time, this dominant factor will lead the technological research and development direction of the whole region by being imitated by neighboring regions.

## 4. Conclusions

We first used Geoda to calculate the Moran's I index and plotted the corresponding Moran scatter plot for spatial correlation analysis. After verifying the spatiality, LM test and joint likelihood ratio LR test were used to estimate the LR and Wald values of the static spatial panel, and the Hausman model was selected. Furthermore, spatial models can be classified as estimates of dynamic Spatial Durbin Model. After partial differential decomposition of the coefficients of the dynamic Spatial Durbin Model, in order to make the estimation more robust, we further used DMSP/OLS night light data and Malmquist productivity index to replace the relevant variables, and utilized this to re-estimate from a more objective perspective. Although the data sample replacing the dependent variable is DMSP/OLS night light data, there is no significant heterogeneity in the estimation results in terms of significance and specific direction of action.

Our conclusion is that: whether using geographic matrices, economic matrices, or fiscal decentralization matrices for verification, local provinces are influenced by tax and fee reduction policies and tend to improve their own independent research and development levels in the short term, while neighboring provinces use them for technology purchase and dependence; Both direct and indirect effects are negative in the long run, indicating that tax and fee reduction policies not only fail to improve the local level of independent research and development in the long run, but also that in neighboring provinces; From a long-term and short-term perspective, local governments have strategic behaviors of political promotion incentives and independent innovation, rather than substantially improving the level of independent innovation; In addition, the externality of policy effects in the region also have a two-way effect of feedback.

In summary, the policy implications proposed in this study are as follows:

First of all, spatial agglomeration requires the government to fully consider the local innovation background and economic basis when implementing policies and to break the resource endowment of administrative divisions. Fiscal tax and fee reduction policies are closely related to the technological preference of the region, and act on the surrounding areas by spillover effect, which has certain externality. However, under the circumstances of interregional economic and political autonomy, administrative regions are still restricted by administrative barriers. Technological progress need manpower and material resources, and in order to realize the national core competitiveness and innovation ability of ascension, When making policies, the government should not only focus on a single province or autonomous region, but should subdivide the economy from the perspective of regional overall planning to ensure the effective flow of elements and resources in the integrated.

Secondly, spatial feedback requires the government to focus on the two-way interaction of policy effects on independent research and development in neighboring regions when making

policies, rather than the one-way assistance or imitation or learning for a province or region. Due to historical reasons such as geography, history, economy and politics, various regions show different economic attributes and characteristics in the process of development. However, due to the existence of externality and feedback effect, independent innovation can fully absorb the surrounding nutrients to make a breakthrough in core competitiveness, so as to formate that a central point drives the regional economy. The policy implications of reducing taxes and fees has a horizontal effect on the incentive effect of independent research and development, and can realize the selection of the advantageous factors of labor and capital for independent research and development in this region. To sum up, feedback effect can be considered to test the level of local independent innovation.

Finally, the spatial time lag requires policy makers not to focus only on the administration period. Local governments generally have a term of administration, which is short-term in the term of office of officials, while independent research and development and the cultivation of core competitiveness is a long-term project. This kind of contradiction easily leads to the performance of independent research and development as strategic motivation. Policy makers should consider long-term development as a goal for assessing officials' tenure. At the same time, the accountability system or the lifetime system should continue to be used to evaluate the innovation performance of officials who leave office, and the behavior of taking no responsibility for the long term should be banned.

It should be noted that our research is different from the innovative development model of the United States and other developed countries. For the United States, as a super developed country, this is the starting point of innovation. As a developing country, to achieve a higher level of independent research and development, we must rely on the advantages of domestic policies and the level of foreign technology absorption and introduction. We have combed the development model of China's innovation practice, which can enrich innovation theory to a certain extent, especially in developing countries. In both theory and practice, our research will help developing countries choose appropriate innovation theories and independent research and development practices, rather than copy the innovation theories and models of developed countries.

## Supporting information

**S1 Dataset.**
(XLS)

**S1 Raw images.**
(ZIP)

## Acknowledgments

We thank the editors and the anonymous reviewers for providing thoughtful and valuable comments.

## Author Contributions

**Conceptualization:** Meng Wu.

**Data curation:** Meng Wu, Ruoyuan Sun.

**Formal analysis:** Meng Wu.

**Funding acquisition:** Meng Wu.

**Methodology:** Meng Wu.

**Writing – original draft:** Meng Wu, Ruoyuan Sun.

**Writing – review & editing:** Meng Wu, Ruoyuan Sun.

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
