## [Editor Report · Decision Letter 0]

6 Dec 2022

PONE-D-22-32394Spatial Spillover Effect of China's Tax and Fee Reduction on Independent Research ——Evidence from dynamic Spatial Dubin analysisPLOS ONE

Dear Dr. Wu,

Thank you for submitting your manuscript to PLOS ONE. After careful consideration, we feel that it has merit but does not fully meet PLOS ONE’s publication criteria as it currently stands. Therefore, we invite you to submit a revised version of the manuscript that addresses the points raised during the review process.

Although you have an interesting and valuable paper, the paper needs to be substantially improved before it can be considered for PLOS ONE. To improve review efficiency and save your time, I first review the paper and provide you some comments. I hope you can consider my comments to improve the paper accordingly to meet the requirements for review. I look forward to receiving your revised version soon.

We look forward to receiving your revised manuscript.

Kind regards,

Baogui Xin, Ph.D.

Academic Editor

PLOS ONE

Journal Requirements:

2. Please upload a new copy of Figure 1 as the detail is not clear. Please follow the link for more information: https://blogs.plos.org/plos/2019/06/looking-good-tips-for-creating-your-plos-figures-graphics/" https://blogs.plos.org/plos/2019/06/looking-good-tips-for-creating-your-plos-figures-graphics/

3. Please clarify the Figures 1 and 2 in your manuscript and separate supporting information S2 and S3 figures file.

Additional Academic Editor Comments:

Dear authors,

This study explores on spatial spillover effect of China's tax and fee reduction on independent research. Although the topic of this research study is interesting and fits within the journal's scope, I think the authors should apply the comments indicated below to increase the quality of the research justification, contributions, and findings.

1. The abstract section is not well written. Authors should simply give the purpose of this paper, and mainly focus on the main findings of this paper.

2. This research is largely framed in the context of China. What about the research findings of other countries?

3. The introduction section does not clearly introduce the research question，the main contributions of this paper and the differences with the other literature. This article lacks a description of innovation

4. What practical/professional and academic consequences will this study have for the future of scientific literature (theoretical contributions)?

Why is this study necessary? Again, the authors should make clear arguments to explain what is the originality and value of the proposed spatial spillover effect of China's tax and fee reduction on independent research in China. This should be stated in the final paragraphs of the introduction and conclusion sections.

In the first and second paragraphs of the introduction, the authors expressed an intention to understand the profile of spatial spillover effect, China's tax and fee reduction, independent research, and innovation performance. Also, bear in mind that contexts matter a lot unless contextual factor is taken into consideration.

The contribution of your work should be better highlighted. The introduction should outline:

(1) What is already known about the topic?

(2) What is not known about the subject and hence what does the study intend to examine. This means outline on what is the gap you seek to fill?

(3) What are the specific research questions the study focuses on?

5. The literature review was inadequate. I advise author to separate the literature review from Section 1. The research gap should be identified based on a more solid foundation.

The literature was not well organized. The literature review was inefficient. It can be organized in terms of different themes /research questions/ theories.

For instance, the author(s) listed many studies in the section of "theoretical background and hypotheses development". It is confusing. What are the exact theories employed in this study? Does the relevant literature correspond to the development of the four hypotheses?

This paper is an inaccurate representation of existing research gap. The literature cited in this paper is older and mostly non-international, which makes it difficult to check the authenticity of the cited ideas. The seminal literature should be cited.

6. Variables are not enough sufficiently supported or clearly explained. So I advise authors to propose some suitable assumptions for your models following some literature review, which can make readers very clear about the source and basis of your assumptions.

7. The methodological contribution is good but they are similar to existing literature. I suggest authors provide explanations on what and why the methodology was applied different and valuable compared to existing methods. Why did not propose…….model. Please more interpretation.

8. I would like to see more discussion of the literature so that I can clearly identify the article relates to competing ideas. The discussion needs to be a coherent and cohesive set of arguments that take us beyond this study in particular and help us see the relevance of what the authors have proposed. The authors need to contextualize the findings in the literature and need to be explicit about the added value of your study towards that literature. Also, other studies should be cited to increase the theoretical background of each of the methods used. Findings should be contextualized in the literature and should be explicit about the added value of the study towards the literature. The contribution and implications of the article are yet to be specified. Please refer the style, 10.1016/j.jclepro.2022.132635

9. Please check your submission in the review system. The Figures and equations are not suitable for publication. Please improve their quality. I advise author rewrite your equations using MathType or other professional equation tools.

Please present more figures and tables to illustrate your results

10. Professional editing is required.

---

## [Author Response · Author response to Decision Letter 0]

31 Jan 2023

Dear reviewers and editors, 

First of all, thank you very much for reading and revising my manuscript. Those comments are all valuable and very helpful for revising and improving our paper, as well as the important guiding significance to our researches. We have studied comments carefully and have made correction which we hope meet with approval. Revised portion are marked in red in the paper. The main corrections in the paper and the responds to the reviewer’s comments are as flowing: 

1.Comment: 

The abstract section is not well written. Authors should simply give the purpose of this paper, and mainly focus on the main findings of this paper.

Response and revise: 

Many thanks to the reviewer for the comment of our abstract. Because some of the words in our article are relatively complex, the purpose and main findings of this study described in the abstract of our manuscript are really difficult to understand, which really causes confusion for reviewers and readers.

Here, we strongly agree with the reviewer. Here, we strongly agree with the reviewer's comment and we have reorganized the abstract content according to the principles requested by the review editor in order to give a simple and concise abstract so that the readers can understand the main purpose of our research and the content of the study. Please refer to the red content in the summary for details. 

2.Comment: 

This research is largely framed in the context of China. What about the research findings of other countries?

Response and revise:

Thanks for the prompt of the reviewer, the description of the research background of tax reduction and fee reduction, innovation and economic development in our manuscript really lacks the description of the background significance of other countries.

After discussion with my tutor, we very much agree with the reviewer's suggestion to add a description of the relevant research background of other countries in the manuscript. Due to space limitation, please see the red letter in the script for specific modifications in the introduction.

3.Comment: 

The introduction section does not clearly introduce the research question，the main contributions of this paper and the differences with the other literature. This article lacks a description of innovation.

Response and revise:

Thanks for the guidance of the reviewer. After reading our manuscript, we found that we paid too much attention to the domestic innovation background. In view of this, we reorganize the introduction and add descriptions of research issues, main contributions, differences from other documents and innovations.

4. Comment:

What practical/professional and academic consequences will this study have for the future of scientific literature (theoretical contributions)?

Why is this study necessary? Again, the authors should make clear arguments to explain what is the originality and value of the proposed spatial spillover effect of China's tax and fee reduction on independent research in China. This should be stated in the final paragraphs of the introduction and conclusion sections.

In the first and second paragraphs of the introduction, the authors expressed an intention to understand the profile of spatial spillover effect, China's tax and fee reduction, independent research, and innovation performance. Also, bear in mind that contexts matter a lot unless contextual factor is taken into consideration.

The contribution of your work should be better highlighted. The introduction should outline:

(1) What is already known about the topic?

(2) What is not known about the subject and hence what does the study intend to examine. This means outline on what is the gap you seek to fill?

(3) What are the specific research questions the study focuses on?

Response and revise:

Thanks again for the detailed suggestions of the reviewer. After discussion with my tutor, we reorganized the literature in the introduction, supplemented the theoretical research, and described the necessity and the background and methods of tax reduction in other countries. On this basis, we summarized the research issues and the gaps filled. Please refer to the introduction of the article for details.

5.Comment:

The literature review was inadequate. I advise author to separate the literature review from Section 1. The research gap should be identified based on a more solid foundation.

The literature was not well organized. The literature review was inefficient. It can be organized in terms of different themes /research questions/ theories.

For instance, the author(s) listed many studies in the section of "theoretical background and hypotheses development". It is confusing. What are the exact theories employed in this study? Does the relevant literature correspond to the development of the four hypotheses?

This paper is an inaccurate representation of existing research gap. The literature cited in this paper is older and mostly non-international, which makes it difficult to check the authenticity of the cited ideas. The seminal literature should be cited.

Response and revise:

Thank the reviewer for reading the second part of our manuscript. We sincerely read the reviewer's suggestions. It may be the mistake in naming the original title of the second part that caused the reviewer to think that the second part is a literature review. In fact, the second part is the derivation of the econometric model. Unlike traditional research, which establish research hypotheses through literature - induction, summary and generalization. With the help of mathematical model, we use the constant effect function of substitution elasticity (CES) nested in Cobb Douglas (CD) function, and further introduce the elements of innovation and tax reduction policy to derive the research hypothesis through mathematical model deduction.

Our four assumptions are derived from the above mathematical model through derivation and differentiation. In addition, because CD production function and other basic mathematical theories are indeed put forward for a long time, on the basis of the mathematical model, we believe that we should still use the previous mathematical basis and introduce relevant elements on the basis of the theorem to derive. So reviewer will feel that the documents are outdated.

In order to avoid misunderstanding between reviewers and readers, we change the chapter name of Part II and briefly introduce the relevant introduction to the beginning of Part II.

6. Comment:

Variables are not enough sufficiently supported or clearly explained. So I advise authors to propose some suitable assumptions for your models following some literature review, which can make readers very clear about the source and basis of your assumptions.

Response and revise:

Thank the reviewer for the preciseness and meticulousness. We describe the variables that are not explained or less explained in strict accordance with the reviewer' suggestions. In addition, due to the problem of model description structure, the reasonable assumptions for the model have been elaborated in fact, but are relatively too complex. To avoid the doubts of readers and reviewer, the article adds a general description of the reasonable assumptions and basis of the model in "2.2 Spatial Durbin Model (SDM) method".

7. Comment:

The methodological contribution is good but they are similar to existing literature. I suggest authors provide explanations on what and why the methodology was applied different and valuable compared to existing methods. Why did not propose…….model. Please more interpretation.

Response and revise:

The reviewer's comments are very pertinent, and we very agree with the reviewer's suggestions. In order to enable readers to better understand the difference between the model used in the article and other models, and to understand the specific contribution and different values of the method, we have added a reasonable explanation of the spatial Dubin model (SDM) method, as well as the purpose and original intention of the method.

8. Comment:

I would like to see more discussion of the literature so that I can clearly identify the article relates to competing ideas. The discussion needs to be a coherent and cohesive set of arguments that take us beyond this study in particular and help us see the relevance of what the authors have proposed. The authors need to contextualize the findings in the literature and need to be explicit about the added value of your study towards that literature. Also, other studies should be cited to increase the theoretical background of each of the methods used. Findings should be contextualized in the literature and should be explicit about the added value of the study towards the literature. The contribution and implications of the article are yet to be specified. Please refer the style, 10.1016/j.jclepro.2022.132635.

Response and revise:

Thank you very much for reviewer’s professionalism. Literature is indeed an indispensable part of the thesis. Because there are a large number of mathematical model derivation in our paper, compared with traditional articles, it really appears that there are few documents. After discussion with our tutor, we tried to reorganize the literature, search and organize relevant research theories again. Limited by length, please refer to the revision of the full paper for details.

9.Comment: 

Please check your submission in the review system. The Figures and equations are not suitable for publication. Please improve their quality. I advise author rewrite your equations using MathType or other professional equation tools.

Please present more figures and tables to illustrate your results.

Response and revise:

It is our negligence that the format problem is not adjusted properly. Thank the reviewer for reminding us. We use tools such as Math Type to adjust the format of the text in strict accordance with the reviewer's comments.

10.Comment: 

Professional editing is required.

Response and revise:

Thank the reviewer for the suggestions. According to the suggestion of the reviewer, we invite our tutor and experts in relevant fields to put forward their opinions again and revise the article.

Finally, thank you again for your guidance, and thank you for reviewing and revising my revised paper again. I hope I can complete an excellent paper with your guidance and help. I sincerely hope my paper can be published in PLOS ONE. 

Sincerely 

Salute!

 Meng Wu, Ruoyuan Sun

---

## [Decision Letter · Decision Letter 1]

27 Feb 2023

PONE-D-22-32394R1

Spatial Spillover Effect of China's Tax and Fee Reduction on Independent Research ——Evidence from dynamic Spatial Dubin analysis

PLOS ONE

Dear Dr. Sun,

Thank you for submitting your manuscript to PLOS ONE. After careful consideration, we feel that it has merit but does not fully meet PLOS ONE’s publication criteria as it currently stands. Therefore, we invite you to submit a revised version of the manuscript that addresses the points raised during the review process.

Although you have an interesting and valuable paper, the paper needs to be substantially improved before it can be considered for PLOS ONE. To improve review efficiency and save your time, I first review the paper and provide you some comments. I hope you can consider my comments to improve the paper accordingly to meet the requirements for review. I look forward to receiving your revised version soon.

We look forward to receiving your revised manuscript.

Kind regards,

Baogui Xin, Ph.D.

Academic Editor

PLOS ONE

Reviewers' comments:

Reviewer's Responses to Questions

**Comments to the Author**

1. If the authors have adequately addressed your comments raised in a previous round of review and you feel that this manuscript is now acceptable for publication, you may indicate that here to bypass the “Comments to the Author” section, enter your conflict of interest statement in the “Confidential to Editor” section, and submit your "Accept" recommendation.

Reviewer #1: (No Response)

Reviewer #2: (No Response)

Reviewer #3: (No Response)

2. Is the manuscript technically sound, and do the data support the conclusions?

Reviewer #1: Yes

Reviewer #2: No

Reviewer #3: Partly

3. Has the statistical analysis been performed appropriately and rigorously? 

Reviewer #1: Yes

Reviewer #2: No

Reviewer #3: Yes

4. Have the authors made all data underlying the findings in their manuscript fully available?

Reviewer #1: No

Reviewer #2: Yes

Reviewer #3: Yes

5. Is the manuscript presented in an intelligible fashion and written in standard English?

Reviewer #1: No

Reviewer #2: No

Reviewer #3: No

6. Review Comments to the Author

Reviewer #1: I strongly agree with the comments given by the previous reviewers. The authors have addressed their concerns in their revision well, and the current version is much better compare to the previous draft. However, I think there are still some prominent issues that need to be addressed.

1.A concise and factual abstract is required. The abstract should state briefly the purpose of the research, the principal results and major conclusions. The abstract still needs to reorganize, especially the main findings of this study.

2.The authors spend a lot of time in the introduction section describing the background of the “innovation”, but do not clearly give the authors' motivation, main work and contribution.

3.The key explanatory variable is questionable in the current draft. This paper use “the general public budget revenue” to measure “tax and fee reduction” is unconvincing. As this is the key explanatory variable, the authors should make 120% effort to cross-evaluate their findings for robust measures on “tax and fee reduction”, otherwise, the analysis does not make sense.

4.There is non-English language in the figures, e.g. Figure 1 and Figure 2. The author should correct them.

5.I would like the authors to polish their current draft towards better writing.

Reviewer #2: This manuscript studied the implementation effect of China's tax and fee reduction policies on independent innovation. However, the manuscript still needs great improvement. I have the honor to review the paper，the modification suggestions are as follows:

1. A large number of professional terms are not used strictly enough, such as line 262, which should be Differences-in-Differences rather than double differences. And “breakpoint regression” “spatial metricis model” (line 270) and “spatial Dubin model” (line 30) are not correct also.

2. The “Literature reviews” part seems weak, which required to improve and strengthen. I suggest to cite more latest researches in the relevant field to provide an up-to-date picture of work.

3. Line 263: Maybe spatial DID model is better. It can not only realize the spatial correlation between variables, but also well evaluate the policy effect.

4. Line 272-273: “Where the spatial weight matrix is added (the explanatory variable or the interpreted variable), it indicates which model this model is.” What does this sentence mean?

5. It is confusing that does not clearly describe each explanatory variable and explained variable as “Section 2.3.1” shows.

6. Have other studies which support using DMSP/OLS night lighting data to describe the deviation of local fiscal competition strategy?

7. Is the variable of “Independent research and development level” logarithmic?

8. Line 447:IS “Spatial diagnostic test” a Spatial autocorrelation test? It should be explained.

9. This paper has constructed three spatial weight matrices. The spatial autocorrelation test did not indicate which matrix was used.

10. Line 471: The picture contains Chinese characters.

11. Since this paper selects the dynamic spatial model in section 3.2, why does the Table 4 not include the coefficient of time lag?

12. It should have regional heterogeneity or economic development heterogeneity analysis and empirical test?

13. In my opinion, adding policy background as a part will make the paper more complete.

14. The conclusion is deficient. Authors should refine the important results and list them.

15. Further, authors should check the grammatical errors. I suggest authors to proof and edit the entire manuscript, and it will significantly help to improve English writing skills.

Reviewer #3: The manuscript researches the implementation effect of tax reduction policies on autonomous innovation in China with the dynamic spatial Durbin model (SDM). Three spatial weight matrices are constructed to conduct extensive empirical research and obtain some results. The following aspects need to be improved.

1.Please check your manuscript and improve its quality. The Figures and equations are not suitable for publication, especially Fig. 1 and Fig. 2.

2. Variables are not enough sufficiently supported or clearly explained in the manuscript. For example, "We tested the implementation effect of China's tax and fee reduction policies on independent innovation with the help of the dynamic spatial Dubin model (SDM), using DMSP/OLS night lighting data and Malmquist productivity index" in the abstract. However, the Malmquist productivity index is not explained in Section 2.3 Variables and data sources. Please explain.

7. PLOS authors have the option to publish the peer review history of their article (what does this mean?). If published, this will include your full peer review and any attached files.

Reviewer #1: No

Reviewer #2: No

Reviewer #3: No

---

## [Author Response · Author response to Decision Letter 1]

12 May 2023

Dear reviewers and editors, 

First of all, thank you very much for reading and revising my manuscript. Those comments are all valuable and very helpful for revising and improving our paper, as well as the important guiding significance to our researches. My supervisor and I have made correction, which we hope meet with approval. Revised portion are marked in the paper. The main corrections in the paper and the responds to the reviewer’s comments are as flowing: 

Reviewer #1:

1.Comment: 

A concise and factual abstract is required. The abstract should state briefly the purpose of the research, the principal results and major conclusions. The abstract still needs to reorganize, especially the main findings of this study.

Response and revise: 

Many thanks to the reviewer for the comment of our abstract. Because some of the words in our article are relatively complex, the purpose and main findings of this study described in the abstract of our manuscript are really difficult to understand, which really causes confusion for reviewers and readers.

Here, we strongly agree with the reviewer's comment and we have reorganized the abstract content according to the principles requested by the review editor in order to give a simple and concise abstract so that the readers can understand the main purpose of our research and the content of the study. Please refer to the red content in the summary for details. 

2.Comment: 

The authors spend a lot of time in the introduction section describing the background of the“innovation”, but do not clearly give the authors' motivation, main work and contribution.

Response and revise:

The reviewer's comments are very pertinent, and the text does tend to confuse the reader in the introduction, which provides less information about the full text. After discussion, we have reorganized the introduction based on the comments of the reviewers, simplified the background of "innovation", and further clarified the motivation, main work, and contributions of the research, so that the reviewers and readers can read it and know exactly what the paper will deal with. 

After the revision, the introduction section of the article includes the domestic and foreign background, relevant research progress, as well as the motivation, main work, and contributions of this article. Due to space limitation, please see the track changes in the script for specific modifications.

3.Comment: 

The key explanatory variable is questionable in the current draft. This paper use “the general public budget revenue” to measure “tax and fee reduction” is unconvincing. As this is the key explanatory variable, the authors should make 120% effort to cross-evaluate their findings for robust measures on “tax and fee reduction”, otherwise, the analysis does not make sense.

Response and revise:

Thanks for the reviewer's suggestion. We may have misunderstood the reviewer and readers due to our unclear description. In fact, we use the change in fiscal revenue as the core indicator for tax and fee reduction. If we rely on previous literature research and choose the changes in certain tax categories or expenses as the indicators for tax and fee reduction, it would be too one-sided, because the tax and fee reduction policy implemented in China is a macro level overall perspective reduction, not a single tax category. So, using the change in fiscal revenue as the core indicator, we can have a clear understanding of the overall momentum of tax and fee reductions. After all, the overall tax and fee reductions are clearly reflected in changes in fiscal revenue, including changes in tax and non tax revenue. Subsequently, we explained the selection range of fiscal revenue indicators in the change of fiscal revenue. Due to the ambiguity of the description, reviewers or readers may mistakenly believe that fiscal revenue is used to define tax and fee reduction.

Based on the above considerations, we have reorganized the description of the selection of core indicators, as detailed in the revised text.

4. Comment:

There is non-English language in the figures, e.g. Figure 1 and Figure 2. The author should correct them.

Response and revise:

Thanks to the reviewers for their rigor and meticulousness, which allowed us to re-examine our manuscript. 

We followed the reviewers' suggestions and reorganized the content of the manuscript to remove non English language expressions.

5.Comment:

I would like the authors to polish their current draft towards better writing.

Response and revise:

Thanks to the reviewer, we have reorganized and polished the manuscript, and have invited our supervisor and relevant supervisors to review it, hoping to further improve our writing skills.Please review the modifications in the article for details.

Reviewer #2:

1.Comment: 

A large number of professional terms are not used strictly enough, such as line 262, which should be Differences-in-Differences rather than double differences. And “breakpoint regression” “spatial metricis model” (line 270) and “spatial Dubin model” (line 30) are not correct also.

Response and revise: 

Thanks for the reviewer's comment. We have reorganized the professional vocabulary in the entire text, especially the professional terminology related to spatial models. At the same time, adjust the capitalization of the terms and unify the writing style. It is worth discussing that in professional terminology, Durbin model can be named Durbin Model or Doberman Model.

For more details, please see the modifications in the text. 

2.Comment: 

The “Literature reviews” part seems weak, which required to improve and strengthen. I suggest to cite more latest researches in the relevant field to provide an up-to-date picture of work.

Response and revise:

The reviewer's comments are very pertinent. It should be noted that we did not write a separate chapter on literature review because China's implementation of tax and fee reductions is relatively short, and the effectiveness of tax and fee reduction policies often lags behind. The scope of policy impact is relatively broad, and most literature is only a partial impact, making it difficult to support a literature review on the entire mechanism of tax and fee reduction effects. We cited relatively new research in the first chapter and inserted relevant research literature into the entire text. Based on the current literature on tax and fee reduction in China, it has been basically covered. Perhaps there are many models in the literature, which has caused a slight hollowing out of the theory. After discussing with our supervisor, we have reorganized the literature and appropriately added some newer literature research.

For details,please see the red letter in the paper.

3.Comment: 

Line 263: Maybe spatial DID model is better. It can not only realize the spatial correlation between variables, but also well evaluate the policy effect.

Response and revise:

The reviewer suggests that using a spatial DID model is indeed a good choice. In fact, before writing our paper, we tried the spatial DID model because it does have natural advantages in policy impact assessment models. Compared with traditional policy impact assessment methods, the DID model can largely avoid endogeneity issues, and setting virtual variables to determine whether policies occur can make estimates more scientific and accurate.

 However, because China's tax and fee reduction policy gradually began after 2008 and has been implemented in some areas, the use of fixed time points to divide the treatment group and control group dummy variables is clearly inaccurate. If a multi-period DID model is adopted, firstly, it is difficult to ensure that the tax and fee reduction policies after 2008 can pass the parallel trend test, because there are many policies implemented in different production fields in China every year, and the tax and fee reduction involves too many fields to completely rule out the correlation effects of policies in various fields; Secondly, when we collect data to form a panel, due to sample data limitations, we are unable to fully cover all fields, and the annual group changes in each field cannot be tracked, making it impossible to use DID to evaluate the overall effect of tax and fee reduction policies; The third requirement is that DID meets exogeneity, and many policies in China provide a large amount of financial subsidies for independent innovation, which may lead to the impact of different policies entering the perturbation term, resulting in biased policy effects estimated by DID.

In addition, we have sorted out the relevant policies and measures for reducing taxes and fees.After the outbreak of the international financial crisis in 2008, China began implementing tax and fee reduction policies. The overall evolution of China's tax and fee reduction policies has gone through three stages:

During the period from 2008 to 2011, in response to the impact of the international financial crisis, especially when the export growth rate was negative 16% in 2019, macroeconomic policies were mainly based on demand management measures to respond to business cycle fluctuations. As far as positive fiscal policies were concerned, the focus was on driving economic growth through large-scale infrastructure construction marked by "four trillion yuan". The tax and fee reduction policies during this period were mainly structural tax reductions.

From 2012 to 2018, the tax and fee reduction policy, as an important component of the supply side structural reform "cost reduction" measures, played an increasingly important role in active fiscal policies. With the implementation of a series of policies such as the pilot and comprehensive implementation of the "business tax to value-added tax" reform, universal fee reduction, halving of income tax for small and micro enterprises, additional deduction of enterprise research and development expenses, and the combination of comprehensive and classified personal income tax, the scale of tax reduction and fee reduction continues to expand, and the macro tax burden level begins to decline.

Since 2019, in order to deal with the trade dispute between China and the United States and the impact of the COVID-19, on the basis of the tax and fee reductions of about 1 trillion yuan and 1.3 trillion yuan in 2017 and 2018, the scale of burden reduction of policies such as the larger scale tax and fee reductions implemented in 2019 and the large-scale rescue and assistance launched against the epidemic in 2020 has reached 2.36 trillion yuan and 2.6 trillion yuan respectively, and the scale of new tax and fee reductions in 2021 is about 1.1 trillion yuan.

In summary, spatial DID does have natural policy evaluation advantages, but neither single period DID nor multi period DID can adapt to the overall effect evaluation of tax and fee reduction policies. Therefore, we chose Spatial Durbin Model.

4. Comment:

Line 272-273: “Where the spatial weight matrix is added (the explanatory variable or the interpreted variable), it indicates which model this model is.” What does this sentence mean?

Response and revise:

Thanks to the reviewer for your careful and meticulous work. It is indeed our negligence that caused confusion for the reviewer and readers. Our original intention is to express that as long as the spatial weight matrix is selected, the form of the Spatial Durbin Model can be determined. There is indeed a problem with what we described in this paragraph.

After discussing with our supervisor, we have reorganized the content of this section and strive to correct any semantic ambiguity. Please see the text for modifications.

5.Comment:

It is confusing that does not clearly describe each explanatory variable and explained variable as “Section 2.3.1” shows.

Response and revise:

According to the suggestions of the reviewer, We do have a relatively simple description of this section, which has caused misunderstandings among the reviewers and readers. After discussing with our supervisor, we sorted out various variables and added descriptions of relevant variables to the paragraph text, such as “The degree of urbanization can represent the overall development level of a city, and can also be directly used to evaluate the economic development level and its derivative can partly represent the growth rate.”“The indicator used to measure the degree of openness of a country or province to the outside world can be foreign direct investment/GDP or import-export volume volume/GDP. We select foreign direct investment/GDP per capita and to prevent values from being difficult to observe, we list the original variables in Table 1”.

6. Comment:

Have other studies which support using DMSP/OLS night lighting data to describe the deviation of local fiscal competition strategy?

Response and revise:

Yes, there is research supporting the use of DMSP/OLS night lighting data to describe deviations in local fiscal competition strategies. There is a lot of research in related fields in China, such as drawing on the research of Zhu Deyun, Sun Ruoyuan, and Wang Bin (2019), considering the bias of local fiscal competition strategies under political promotion incentives, and referring to Liu Yanxu, Wu Wenheng, Wen Xiaojin, and Zhang Donghai (2013) using DMSP/OLS night light data to describe the bias of local fiscal competition strategies. Nighttime lighting data can to some extent display the construction of local infrastructure, which can reveal the competitive behavior of local governments towards infrastructure.

7. Comment:

Is the variable of “Independent research and development level” logarithmic?

Response and revise:

This is our negligence, as the description can easily lead to misunderstandings.Our original intention was to display the logarithmic data of the control variables. The previous reviewer asked us to display the data of the explained variable and the explanatory variable in Table 1 together. However, we overlooked that the control variables in the table were logarithmic and directly included the original data. Thanks to the reviewer for your careful and meticulous work. The logarithmic form of all variables used in the calculation of our article's model, so we have adjusted the data display format in the table to take logarithmic form.

8. Comment:

Line 447:IS “Spatial diagnostic test” a Spatial autocorrelation test? It should be explained.

Response and revise:

Spatial diagnostic test is a spatial autocorrelation test in our manuscript. Thanks for the reviewer's suggestion. The doubts of the reviewer are also the doubts of the readers. After discussing with our supervisor, we have changed the title of the chapter to "Spatial autocorrelation test" and added explanations on spatial autocorrelation testing. “Spatial autocorrelation statistics are a fundamental property used to measure geographic data: the degree of interdependence between data at a certain location and data at other locations. Usually, this dependency is called spatial dependency. Due to the influence of spatial interaction and diffusion, geographic data may no longer be independent of each other, but rather related. Spatial autocorrelation is an important starting point for the setting of spatial models. ”

9.Comment: 

This paper has constructed three spatial weight matrices. The spatial autocorrelation test did not indicate which matrix was used.

Response and revise:

Thanks for the reviewer's suggestion. We supplement the description of the spatial weight matrix in the article."we use the Moran's I index for spatial autocorrelation test, where the spatial weight matrix selects the spatial adjacency matrix (0-1 matrix). The spatial weight of adjacent provinces is 1, and the spatial weight of non adjacent provinces is 0. The Moran's I index is an indicator used to measure spatial correlation, and its core is the spatial adjacency matrix".

10.Comment: 

Line 471: The picture contains Chinese characters.

Response and revise:

Thanks to the reviewer for your diligence and attention. We changed Chinese characters to English according to the comments of the reviewer.

11.Comment: 

Since this paper selects the dynamic spatial model in section 3.2, why does the Table 4 not include the coefficient of time lag?

Response and revise:

Thank the reviewer for the suggestions. We include spatial lag coefficients, and the spatial lag coefficients of the dependent variable are three: "time lag items of independent research and development", "W * independent research and development level", "W * time lag of independent research and development". W * policy effect is the product of four spatial weights and explanatory variables respectively. The meaning of spatial lag is different from that of time lag, and it is the impact of the surrounding area on the research area. In fact, this influence is the representation of the spatial weight matrix. In the dynamic spatial model, we selected four spatial weight matrices. Unlike what was estimated by the static spatial model, the static spatial model did not distinguish spatial weights, so we noted the spatial lag term, which is included in the W space weight. In addition, the theme of our manuscript does not involve the time lag of explanatory and control variables, as the time lag terms of explanatory and control variables do not affect the theme of this article. But the dynamic spatial model is the time lag of the dependent variable, which we have already included. We want to present the description of the variables as clearly as possible, as the characters may be too long, causing misunderstandings among the reviewer and readers.

12.Comment: 

It should have regional heterogeneity or economic development heterogeneity analysis and empirical test?

Response and revise:

The reviewer studied the measurement method in depth and paid more attention to the robustness test of the estimator. In fact, in our manuscript, the spatial model adopts both static and dynamic models, including the selection of four spatial weights, which can already test the heterogeneity of regions and economies. Because adjacency matrix and geographical location matrix can verify regional heterogeneity, economic matrix and fiscal decentralization matrix are actually the embodiment of economic development heterogeneity. In addition, the introduction of light data is also a test of economic development heterogeneity to some extent.We have constructed a static model and a dynamic spatial model. The introduction of spatial and temporal parameters, as well as the spatial and time lag terms of the explained variables can overcome some endogeneity to a certain extent, and the decomposition of the coefficients of the Dynamic Spatial Durbin Model can more purely determine the marginal utility of the parameters, which has good persuasiveness for the subsequent estimation results of this paper. Furthermore, different spatial weight types were constructed, such as the adjacency weight matrix, geographical weight matrix, economic weight matrix, and fiscal decentralization weight matrix, which can improve the explanatory power of the parameters.

Perhaps our explanation of this section is not very clear, so we reorganized and explained the above question in the text to avoid confusion among reviewer and readers.

13.Comment: 

In my opinion, adding policy background as a part will make the paper more complete.

Response and revise:

Thanks to the reviewer for the suggestion. As the reviewers pointed out, our article lacks a description of the policy background, which is indeed a bit abrupt. After discussing with our supervisor, we have added the policy background of tax reduction and fee reduction in the introduction. Due to space limitations, please see the modifications in the text for details. 

“We have sorted out the relevant policies and measures for reducing taxes and fees. The overall evolution of China's tax and fee reduction policies has gone through three stages: During the period from 2008 to 2011, in response to the impact of the international financial crisis, especially when the export growth rate was negative 16% in 2019, macroeconomic policies were mainly based on demand management measures to respond to business cycle fluctuations. The tax and fee reduction policies during this period were mainly structural tax reductions; From 2012 to 2018, the tax and fee reduction policy, as an important component of the supply side structural reform "cost reduction" measures, played an increasingly important role in active fiscal policies. With the implementation of a series of policies such as the pilot and comprehensive implementation of the "business tax to value-added tax" reform, universal fee reduction, halving of income tax for small and micro enterprises, additional deduction of enterprise research and development expenses, and the combination of comprehensive and classified personal income tax, the scale of tax reduction and fee reduction continues to expand, and the macro tax burden level begins to decline; Since 2019, in order to deal with the trade dispute between China and the United States and the impact of the COVID-19, on the basis of the tax and fee reductions of about 1 trillion yuan and 1.3 trillion yuan in 2017 and 2018, the scale of burden reduction of policies such as the larger scale tax and fee reductions implemented in 2019 and the large-scale rescue and assistance launched against the epidemic in 2020 has reached 2.36 trillion yuan and 2.6 trillion yuan respectively, and the scale of new tax and fee reductions in 2021 is about 1.1 trillion yuan”.

14.Comment:

The conclusion is deficient. Authors should refine the important results and list them.

Response and revise:

Thanks to the reviewer for the insightful suggestions. We have sorted out the relevant writing process and conclusions based on the comments of the reviewers:

We first used Geoda to calculate the Moran's I index and plotted the corresponding Moran scatter plot for spatial correlation analysis. After verifying the spatiality, LM test and joint likelihood ratio LR test were used to estimate the LR and Wald values of the static spatial panel, and the Hausman model was selected. Furthermore, spatial models can be classified as estimates of Dynamic Spatial Durbin model. After partial differential decomposition of the coefficients of the Dynamic Spatial Durbin model, in order to make the estimation more robust, we further uses DMSP/OLS night light data and Malmquist productivity index to replace the relevant variables, and uses this to re-estimate from a more objective perspective. Although the data sample replacing the dependent variable is DMSP/OLS night light data, there is no significant heterogeneity in the estimation results in terms of significance and specific direction of action.

Our conclusion is that: whether using geographic matrices, economic matrices, or fiscal decentralization matrices for verification, local provinces are influenced by tax and fee reduction policies and tend to improve their own independent research and development levels in the short term, while neighboring provinces use them for technology purchase and dependence; Both direct and indirect effects are negative in the long run, indicating that tax and fee reduction policies not only fail to improve the local level of independent research and development in the long run, but also that in neighboring provinces; From a long-term and short-term perspective, local governments have strategic behaviors of political promotion incentives and independent innovation, rather than substantially improving the level of independent innovation; In addition, the externality of policy effects in the region also have a two-way effect of feedback.

15.Comment:

Further, authors should check the grammatical errors. I suggest authors to proof and edit the entire manuscript, and it will significantly help to improve English writing skills.

Response and revise:

Thanks for the reviewer's comment. After discussion with my supervisor, we combed the manuscript again, checked syntax error, and proofread the entire manuscript in strict accordance with the comments of the reviewer. We hope to improve our English writing skills through repeated revisions. Please see the red characters in the text for detailed modifications.

Reviewer #3:

1. Comment:

Please check your manuscript and improve its quality. The Figures and equations are not suitable for publication, especially Fig. 1 and Fig. 2.

Response and revise:

Thanks for the suggestions from the reviewer. We have strictly followed the reviewer’s advice and adjusted the figures and equations in the manuscript. Please see the manuscript for detailed revisions.

2. Comment:

Variables are not enough sufficiently supported or clearly explained in the manuscript. For example, "We tested the implementation effect of China's tax and fee reduction policies on independent innovation with the help of the dynamic spatial Dubin model (SDM), using DMSP/OLS night lighting data and Malmquist productivity index" in the abstract. However, the Malmquist productivity index is not explained in Section 2.3 Variables and data sources. Please explain.

Response and revise:

Thanks for the reviewer's suggestions. After discussing with our supervisor, we strictly followed the reviewer's suggestions and provided additional support and explanations in the manuscript. For the Malmquist productivity index, we add: “The Malmquist index was initially proposed by Malmquist in 1953, and Caves, Christensen, and Diewert began applying this index to the measurement of changes in production efficiency in 1982. The malquist index is a measure of the output input ratio, used to assess production efficiency and can be decomposed into several sub efficiency indicators. The calculation method of Malmquist index can be expressed by a formula: Malmquist index can be decomposed into technical progress index and technical efficiency change index. Among them, the technological progress index is the ratio of productivity indices at two time points, and the technological efficiency change index is the ratio of productivity indices at two time points divided by the technological progress index. The malmquist technology progress index is selected in this manuscript, and the specific calculation is: ”.

Finally, thank you again for your guidance, and thank you for reviewing and revising my revised paper again. I hope I can complete an excellent paper with your guidance and help. I sincerely hope my paper can be published in PLOS ONE. 

Sincerely 

Salute!

Wu Meng, Sun Ruoyuan

---

## [Decision Letter · Decision Letter 2]

1 Jun 2023

PONE-D-22-32394R2Spatial Spillover Effect of China's Tax and Fee Reduction on Independent Research —— Evidence from dynamic Spatial Durbin analysisPLOS ONE

Dear Dr. Sun,

Thank you for submitting your manuscript to PLOS ONE. After careful consideration, we feel that it has merit but does not fully meet PLOS ONE’s publication criteria as it currently stands. Therefore, we invite you to submit a revised version of the manuscript that addresses the points raised during the review process.

We recommend that it should be revised taking into account the changes requested by the reviewers. Since the requested changes includes Minor Revision, the revised manuscript will undergo the next round of review by the same reviewers or only by the Academic Editor.

We look forward to receiving your revised manuscript.

Kind regards,

Baogui Xin, Ph.D.

Academic Editor

PLOS ONE

Journal Requirements:

Reviewers' comments:

Reviewer's Responses to Questions

**Comments to the Author**

1. If the authors have adequately addressed your comments raised in a previous round of review and you feel that this manuscript is now acceptable for publication, you may indicate that here to bypass the “Comments to the Author” section, enter your conflict of interest statement in the “Confidential to Editor” section, and submit your "Accept" recommendation.

Reviewer #1: All comments have been addressed

Reviewer #2: (No Response)

Reviewer #3: All comments have been addressed

2. Is the manuscript technically sound, and do the data support the conclusions?

Reviewer #1: Yes

Reviewer #2: Yes

Reviewer #3: Yes

3. Has the statistical analysis been performed appropriately and rigorously? 

Reviewer #1: Yes

Reviewer #2: Yes

Reviewer #3: Yes

4. Have the authors made all data underlying the findings in their manuscript fully available?

Reviewer #1: Yes

Reviewer #2: Yes

Reviewer #3: Yes

5. Is the manuscript presented in an intelligible fashion and written in standard English?

Reviewer #1: No

Reviewer #2: Yes

Reviewer #3: Yes

6. Review Comments to the Author

Reviewer #1: (No Response)

Reviewer #2: The author(s) revised most of the recommendations, but the following deficiencies remain.

1. The explanations of the explained and explanatory variables are confusing.

2. Figure 1 is not clear. Can it be replaced by a table? Or streamline it to a few years.

3. Lack of description of the latest research trends.

4. The author(s) do not adequately explain the reasons for not using spatial DID.

Reviewer #3: (No Response)

7. PLOS authors have the option to publish the peer review history of their article (what does this mean?). If published, this will include your full peer review and any attached files.

Reviewer #1: No

Reviewer #2: No

Reviewer #3: No

---

## [Author Response · Author response to Decision Letter 2]

27 Jun 2023

Dear reviewers and editors, 

Thanks very much for the comments from the editor and reviewers. We have organized the description and grammar of the manuscript to improve our English writing skills.The main corrections in the paper and the responds to the reviewer’s comments are as flowing: 

Reviewer #2:

1.Comment: 

The explanations of the explained and explanatory variables are confusing.

Response and revise: 

Many thanks to the reviewer for the comment. We originally intended to provide a detailed explanation of the variables of taxes and fees reduction, as well as the alternative lighting data variables used in robustness testing. The comment of the reviewer were indeed very relevant, and we also felt that the language expression was redundant and complex, which was confusing. Therefore, we reorganize the expression, remove redundant descriptions, and simplify the explanations of the dependent and explanatory variables as much as possible. Please refer to the revised manuscript for details. 

2.Comment: 

Figure 1 is not clear. Can it be replaced by a table? Or streamline it to a few years.

Response and revise:

Thanks to the reviewer for the suggestion. Indeed, the Moran's I Index image, which spans 22 years, cannot meet the clarity requirements with pixels. After discussing with our supervisor, we have presented the Moran's I Index chart for 4 years in the article, with images from other years attached. Please refer to the revised manuscript for details.

3.Comment: 

Lack of description of the latest research trends.

Response and revise:

Thanks for the reviewer's suggestion. We strictly follow the reviewer's comment and provide a description of the latest research trends. Please refer to the revised manuscript for details.

4. Comment:

The author(s) do not adequately explain the reasons for not using spatial DID.

Response and revise:

Thanks to the reviewers for their rigor and meticulousness, which allowed us to re-examine our manuscript. 

As suggested by the reviewer, using a spatial DID model is indeed a good choice，because spatial DID does have natural policy evaluation advantages. But neither single-period DID nor multi-period DID can adapt to the overall taxes and fees reduction policy effect evaluation:

1.The single period DID cannot determine the grouping time point between the treatment group and the control group. China's taxes and fees reduction policies include numerous small policies, and the implementation time varies in different regions. The taxes and fees reduction policy gradually began after 2008, and pilot projects were adopted first, with some areas gradually expanding to the whole country. The time points for implementing taxes and fees reduction policies in different provinces are different, and the use of fixed time points to divide processing and control groups into dummy variables is clearly inaccurate.

2.If using a multi-period DID model:

(1) It is impossible to exclude the relevant effects of other policies promoting production. The taxes and fees reduction policy is difficult to pass the parallel trend test of DID, as there are various forms of production promotion policies in different provinces and fields. The taxes and fees reduction policy is only one of the policy means to promote production, and its coverage is wide, making it impossible to completely rule out the correlation effects of different policies;

(2) The exogenous nature of taxes and fees reduction policies cannot be guaranteed. DID requires meeting policy externalities, and apart from tax and fee reduction policies, many other policies in China have a significant impact on independent innovation, which may lead to the impact of different policies entering the perturbation term, leading to a two-way causal relationship, resulting in biased policy effects estimated by DID；

(3) The sample changes rapidly and cannot track all fields, resulting in the inability to use DID to evaluate the overall taxes and fees reduction policy effect. Due to the limitations of sample data and the inability to track the changes in grouping in various fields each year, for example, some cities have implemented tax cuts in the manufacturing industry this year, announced tax cuts in the education industry next year, and some regions have reduced taxes this year but not next year, making it impossible to use DID to evaluate the overall tax and fee reduction policy effect.

Finally, thank you again for your guidance, and thank you for reviewing and revising my revised paper again. I hope I can complete an excellent paper with your guidance and help. I sincerely hope my paper can be published in PLOS ONE. 

Sincerely 

Salute!

Wu Meng, Sun Ruoyuan

---

## [Editor Report · Decision Letter 3]

29 Jun 2023

Spatial Spillover Effect of China's Tax and Fee Reduction Policies on Independent Research and Development —— Evidence from dynamic Spatial Dubin analysis

PONE-D-22-32394R3

Dear Dr. Sun,

We’re pleased to inform you that your manuscript has been judged scientifically suitable for publication and will be formally accepted for publication once it meets all outstanding technical requirements.

Kind regards,

Baogui Xin, Ph.D.

Academic Editor

PLOS ONE
---

## [Editor Report · Acceptance letter]

20 Jul 2023

PONE-D-22-32394R3 

Spatial Spillover Effect of China's Tax and Fee Reduction Policies on Independent Research and Development     Evidence from dynamic Spatial Dubin analysis 

Dear Dr. Sun:

I'm pleased to inform you that your manuscript has been deemed suitable for publication in PLOS ONE. Congratulations! Your manuscript is now with our production department. 

Kind regards, 

on behalf of

Professor Baogui Xin 

Academic Editor

PLOS ONE